# An empirical evaluation of bias correction methods for palaeoclimate simulations

Robert Beyer[1], Mario Krapp[1], and Andrea Manica[1]

[1]Department of Zoology, University of Cambridge, Downing Street, Cambridge CB2 3EJ, United Kingdom

**Correspondence:** Robert Beyer (rb792@cam.ac.uk)

**Abstract.**

Even the most sophisticated global climate models are known to have significant biases in the way they simulate the climate system. Correcting model biases is therefore an essential step toward realistic palaeo-climatologies, which are important for many applications, such as modelling long-term ecological dynamics.

5 Here, we evaluate three widely-used bias correction methods – the Delta Method, Generalised Additive Models (GAMs) and Quantile Mapping – against a large global dataset of empirical temperature and precipitation records from the present, the Mid-Holocene (~6,000 years BP), the Last Glacial Maximum (~21,000 years BP) and the Last Interglacial Period (~125,000 years BP). In most cases, the differences between the bias reductions achieved by the three methods are small. Overall, the Delta Method performs slightly better, albeit not always

10 to a statistically significant degree, at minimising the median absolute bias between empirical data and debiased simulations for both temperature and precipitation than GAMs and Quantile Mapping; however, there is considerable spatial and temporal variation in the performance of each of the three methods. Our data also indicate that it could soon be possible to use empirical reconstructions of past climatic conditions not only for the evaluation of bias correction methods, but for fitting statistical relationships between empirical and simulated data through

15 time that can inform more effective bias correction methods.

## 1 Introduction

Realistic reconstructions of global palaeoclimate are a key input for modelling many important long-term and large-scale ecological processes (Eriksson et al., 2012; Timmermann and Friedrich, 2016; Leonardi et al., 2018; Zhu et al., 2018; Rangel et al., 2018; Beyer et al., 2020). In many of these applications, climatological normals at

quasi-equilibrium of variables such as temperature and precipitation at different points in time represent the most relevant climatic inputs. Simulations of these variables remain subject to substantial biases when compared to observational data, despite advancements in how complex physical processes are represented in global climate models (Solomon et al., 2007; Ehret et al., 2012). Depending on the region of interest, these biases can be of the order of several degrees of temperature, and tens of percent of precipitation, which can make the difference between markedly different vegetation types (Kottek et al., 2006).

Bias correction has received a great deal of attention for present-day and near-future simulations (Ho et al., 2011; Maraun and Widmann, 2018), whereas work on palaeoclimate simulations has been much more limited. This is partly due to the different time scale of palaeoclimatological applications, for which computationally intensive bias correction methods that are used for the recent past and near future are not suitable. Three main methods have been applied thus far to bias-correct climatological normals in the palaeocontext: the Delta Method (http://www.worldclim.org/downscaling, http://www.paleoclim.org/methods/, Armstrong et al., 2019), statistical methods based on generalised additive models (GAMs) (Vrac et al., 2007; Levavasseur et al., 2011; Woillez et al., 2014; Latombe et al., 2018) and Quantile Mapping (Lorenz et al., 2016). The Delta Method assumes that biases are location-specific and constant over time; it uses a map of the local differences between observed and simulated values at present-day to bias-correct past simulations (Maraun and Widmann, 2018). GAMs attempt to represent statistical relationships between present-day simulated climate variables (as well as other known physical variables, such as elevation and the distance from the coast) and present-day observed climate, and apply these relationships to past simulations to reduce biases (Vrac et al., 2007; Maraun and Widmann, 2018). Quantile Mapping adjusts the cumulative distribution of the simulated data by applying a transformation between the quantiles of present-day simulated and observed climate to the quantiles of past simulated climate. (Maraun and Widmann, 2018).

Here we combine a set of high-resolution simulations of the climatological means of several temperature and precipitation variables for the present, the Mid-Holocene (~6,000 years BP), the Last Glacial Maximum (~21,000 years BP) and the Last Interglacial Period (~125,000 years BP) with a global dataset of empirical climatic reconstructions to evaluate the performance of the Delta Method, a GAM-based approach, and Quantile Mapping in removing simulation biases. We focus on the global performance of the different methods, but point out that bias correction is generally not a one-size-fits-all approach (Maraun and Widmann, 2018), and that our

results do not remove the need for local re-evaluations of methods in specific continental and subcontinental regions of interest.

Section 2 provides details of the three bias correction methods, the climate simulations, and the empirical palaeoclimate reconstructions used in this study. In section 3, we quantitatively assess the performance of the methods at the global scale and with regard to spatial and temporal heterogeneities. Section 4 discusses how empircal palaeoclimate reconstructions could be used not only to evaluate methods, but to help estimate the variation of local model biases over time, thus combining the strengths of the Delta Method and statistical bias correction.

## 2  Material and Methods

### 2.1  Climate data

#### 2.1.1  Modelled climate data

We used $1.25° \times 0.83°$ resolution palaeoclimate simulations of monthly mean temperature and monthly precipitation for the present, the Mid-Holocene and the Last Glacial Maximum (LGM) from the HadAM3H atmospheric model (Hudson and Jones, 2002; Arnell et al., 2003), which is part of the family of HadCM3 climate models (Valdes et al., 2017). For the Last Interglacial Period, we do not have simulation data from HadAM3H, but we used the global climate model emulator GCMET (Krapp et al., 2019) that is based on the same model and can make predictions at the same spatial resolution. In all cases, simulations represent climatological normals (i.e. 30-year averages) at quasi-equilibrium, following a 500-year spin-up period. Based on the monthly data, we computed the following climate variables, for which suitable empirical reconstructions are available (see section 2.1.2): terrestrial mean temperature, marine mean annual temperature, temperature of the coldest month, temperature of the warmest month, and annual precipitation. We note that the results presented here may be specific to the particular climate simulations considered, and do not claim generalisability to other models.

Empirical reconstructions (see section 2.1.2) of terrestrial temperature variables were compared against simulated temperature at 1.5 meters height, while simulated air surface temperature was used as a proxy for sea surface temperature, as sea surface temperature is not part of the HadAM3H output. We removed marine data

points for which simulated air surface temperature was below the freezing point of saltwater, −1.8°C, as in this case the simulated value corresponds to the temperature of an ice layer rather than that of the top layer of water.

### 2.1.2 Empirical climate data

All bias correction methods considered here are calibrated based on present-day observational data. For this, we used monthly terrestrial temperature and precipitation data at a 0.167° grid resolution (New et al., 2002), and mean annual sea surface temperature at a 1° grid resolution (Reynolds et al., 2002), representative of 1960–1990. These maps were remapped to the 1.25°×0.83° grid of the palaeoclimate simulations by taking the average of values contained in each target grid cell.

We used global datasets of local empirical palaeoclimatic reconstructions of terrestrial mean annual temperature, temperature of the coldest and warmest month, and annual precipitation, for the Mid-Holocene and the LGM from Bartlein et al. (2011), reconstructions of mean annual sea surface temperature for the Mid-Holocene and the LGM from Hessler et al. (2014) and Waelbroeck et al. (2009), respectively, and reconstructions of mean annual terrestrial and sea surface temperature for the Last Interglacial Period from Turney and Jones (2010). Standard errors of reconstructed values are available for all variables with the exception of terrestrial and marine temperature during the Last Interglacial Period. Terrestrial temperature and precipitation reconstructions for the Mid-Holocene and the LGM are available on a 2° resolution grid, and LGM marine temperature reconstructions are provided on a 5° grid. We assigned each sample of these datasets to the 1.25°×0.8° grid cell of our palaeoclimate simulations (see section 2.1.1) that contains the centre of the relevant 2° or 5° cell. Reconstructions for the Last Interglacial Period are not gridded, and were assigned to the 1.25°×0.8° grid cell that contains the sample location. Figs. 3 and 4 visualise the locations of all empirical reconstructions of terrestrial and marine mean annual temperature, and annual precipitation.

Empirically derived climate reconstructions can themselves be subject to biases and uncertainties, which arise at the different stages of the reconstruction process, from collecting the data to computationally converting empirical records to climatic variables. Nonetheless, these data represent the best empirically-based estimates of past climatic conditions available, and the most suitable data for our analysis.

## 2.2 Bias correction methods

### 2.2.1 The Delta Method

The Delta Method consists of adding the difference between past and present-day simulated climate (the Delta) to present-day observed climate. As such, the Delta Method assumes that local (i.e. grid cell-specific) model biases are constant over time (Maraun and Widmann, 2018). For temperature variables (including terrestrial and marine mean annual temperature, and terrestrial temperature of the warmest and coldest month, considered here), the bias in a geographical location $x$ is given by the difference between present-day observed and simulated temperature, $T_{\mathrm{emp}}(x,0) - T_{\mathrm{sim}}^{raw}(x,0)$. Bias-corrected temperature in $x$ at some time $t$ in the past is estimated as

$$T_{\mathrm{sim}}^{DM}(x,t) := T_{\mathrm{emp}}(x,0) + \left(T_{\mathrm{sim}}^{raw}(x,t) - T_{\mathrm{sim}}^{raw}(x,0)\right)$$
$$= T_{\mathrm{sim}}^{raw}(x,t) + \left(T_{\mathrm{emp}}(x,0) - T_{\mathrm{sim}}^{raw}(x,0)\right). \tag{1}$$

The second expression illustrates that $T_{\mathrm{sim}}^{DM}(x,t)$ is alternatively given by adding the local present-day bias to the local temperature simulated for time $t$.

Precipitation is bounded below by zero and covers different orders of magnitude across different regions. A multiplicative rather than additive bias correction is therefore more common when applying the Delta Method for precipitation, which corresponds to applying the simulated relative change to the observations (Maraun and Widmann, 2018). Analogously to temperature, debiased precipitation is estimated as

$$P_{\mathrm{sim}}^{DM}(x,t) := P_{\mathrm{obs}}(x,0) \cdot \frac{P_{\mathrm{sim}}^{raw}(x,t)}{P_{\mathrm{sim}}^{raw}(x,0)}$$
$$= P_{\mathrm{sim}}^{raw}(x,t) \cdot \frac{P_{\mathrm{obs}}(x,0)}{P_{\mathrm{sim}}^{raw}(x,0)}. \tag{2}$$

### 2.2.2 Statistical Models / GAMs

Statistical bias correction methods assume the existence of a functional relationship between true climatic conditions (dependent variables), and climate model outputs as well as additional known forcings such as topography (independent variables) (Vrac et al., 2007; Maraun and Widmann, 2018). Transfer functions representing this relationship are calibrated based on present-day simulated and observed climate, and are then applied to simulations of past climate to derive bias-corrected data. Generalised additive models (GAMs) have gained particular

popularity as transfer functions (Vrac et al., 2007; Levavasseur et al., 2011; Woillez et al., 2014; Latombe et al., 2018). They accommodate potential nonlinearities in the response of the individual predictor variables, but – owing to the computational requirements of general high-dimensional nonlinear regressions – assume that the interactions between predictors can be neglected.

For a set of geographical locations $x_1, x_2, \ldots$, we denote by $V_{\text{emp}}(x_i, 0)$ the present-day observed value of a climate variable $V$ (representing the relevant temperature and precipitation variables considered here) in the location $x_i$. Here, the $x_1, x_2, \ldots$ represent the locations of the cells of the $1.25° \times 0.8°$ grid of the climate data (see section 2.1) on land and in the ocean in the case of terrestrial and marine climate variables, respectively. In a GAM, the present-day observed values of $V$ are modelled as the sum of functions of variables that are

available both for the present and the past, such as climate model outputs (typically including the raw simulated data of the variable in question), and/or certain geographical or physical quantities that are known across time. We denote the values of these predictor variables in the location $x_i$ at time $t$ by $X_1^V(x_i, t), X_2^V(x_i, t), \ldots$. The $X_j^V$ are generally time-dependent, not only when they are climate model outputs, but also when they represent elevation or the distance to the coast, which vary over time as the result of sea level changes. The GAM is

defined by the regression

$$V_{\text{emp}}(\cdot, 0) \sim \sum_j f_j\big(X_j^V(\cdot, 0)\big), \tag{3}$$

where the $f_1, f_2, \ldots$ represent smooth functions that are fitted to minimise the distance between the left and the right hand side in Eq. (3). Once the model has been calibrated on the present-day data, it is used to estimate the bias-corrected values of the climate variable $V$ in the location $x_i$ at a point $t$ in the past as

$$V_{\text{sim}}^{GAM}(x_i, t) := \sum_j f_j\big(X_j^V(x_i, t)\big). \tag{4}$$

Similar to Latombe et al. (2018), here, we used elevation, the shortest distance to the ocean and simulated temperature as predictor variables $X_j^V$ for temperature variables. Elevation, the shortest distance to the ocean, and simulated annual precipitation, temperature, (absolute) wind speed, air pressure and relative humidity were used as predictors variables for annual precipitation. The functions $f_i$ were estimated as piecewise third order

polynomials (using thin plate splines did not change the results) using the *mgcv* package in R (Wood, 2004).

### 2.2.3 Quantile Mapping

Quantile Mapping aims to correct distributional biases in the simulated climate data. The method consists of first computing a transformation that maps the quantiles of the cumulative distribution function of all present-day observed values (i.e. from all land or ocean grid cells) of a climate variable onto the quantiles of the cumulative distribution function of all present-day simulated values. The derived mapping is then applied to the cumulative distribution function of all simulated values at a given point in the past. For example, let the cumulative distribution function of the values of present-day observed terrestrial mean annual temperature (i.e. from all land grid cells) map the value $T_1^\circ$C onto the value $q \in [0,1]$, and let the analogous cumulative distribution function of present-day simulated terrestrial mean annual temperature map $T_2^\circ$C onto $q$. If the value that is mapped onto $q$ by the cumulative distribution function of simulated terrestrial mean annual temperature at a given point in the past is $T_3^\circ$C, then the bias-corrected mean annual temperature in all grid cells with simulated mean annual temperature $T_3^\circ$C at that point in time is estimated as $T_3 + (T_1 - T_2)^\circ$C. Notably, by design of the method, the cumulative distribution function of present-day bias-corrected simulated data (i.e. after applying Quantile Mapping) is identical to the cumulative distribution functions of present-day observed values.

Formally, denote by $x_1, x_2, \ldots$ the centres of the $1.25^\circ \times 0.8^\circ$ grid cells of the climate data (see section 2.1) on land and in the ocean in the case of terrestrial and marine climate variables, respectively. For a climate variable $V$ (representing the relevant temperature and precipitation variables), we denote by $F_{\text{emp}}^V[0]$ the cumulative distribution function of all present-day empirical observations, $V_{\text{emp}}(x_1, 0), V_{\text{emp}}(x_2, 0), \ldots$ (i.e. $F_{\text{emp}}^V[0]$ is the function that monotonically maps these values onto the interval $[0,1]$). Analogously, we denote by $F_{\text{sim}}^{V,raw}[t]$ the cumulative distribution function of the raw simulated values $V_{\text{sim}}^{raw}(x_1, t), V_{\text{emp}}^{raw}(x_2, t), \ldots$ at time $t$. We denote by $F_{\text{emp}}^V[0]^{-1}$ and $F_{\text{sim}}^{V,raw}[t]^{-1}$ (both mapping $[0,1]$ to $\mathbb{R}$) the inverse functions of $F_{\text{emp}}^V[0]$ and $F_{\text{sim}}^{V,raw}[t]$, respectively. With this notation, $F_{\text{sim}}^{V,raw}[t](V_{\text{sim}}^{raw}(x_i, t))$ is the quantile corresponding to the value $V_{\text{sim}}^{raw}(x_i, t)$ in the set of all simulated values of the climate variable $V$ at time $t$. Under Quantile Mapping, the function $F_{\text{emp}}^V[0]^{-1} - F_{\text{sim}}^{V,raw}[0]^{-1}$ maps each such quantile to a quantile-specific correction term, which is then applied to the raw simulation data. Thus, the bias-corrected value of $V$ in the location $x_i$ at time $t$ is estimated as

$$V_{\text{sim}}^{QM}(x_i, t) := V_{\text{sim}}^{raw}(x_i, t) + \underbrace{\left[ F_{\text{emp}}^V[0]^{-1} - F_{\text{sim}}^{V,raw}[0]^{-1} \right] (F_{\text{sim}}^{V,raw}[t](V_{\text{sim}}^{raw}(x_i, t)))}_{\text{Correction term specific to the quantile of the value } V_{\text{sim}}^{raw}(x_i, t)} . \tag{5}$$

### 2.2.4 Method discussion

All three bias correction methods considered here aim at minimising biases in past simulated data, but they are based on different assumptions as to how this aim can best be achieved. The Delta Method assumes that the known present-day model bias is also a good estimate for past model bias. GAM methods and Quantile Mapping operate on the premise that this assumption of that Delta Method – local biases remaining constant over time – is too strong. Instead, GAM methods assume that a better estimate of past model biases can be obtained by deriving a statistical relationship between present-day bias and present-day simulations, and then applying this relationship to past simulations in order to estimate past bias. Because regressions generally do not fit the data perfectly, present-day biases modelled by the GAM will not exactly match the observed biases across all grid cells. Unlike in the case of the Delta Method, GAM-corrected present-day simulations are therefore not identical to the present-day observed climate. This drawback is accepted under the assumption that the derived statistical model captures the mechanisms that underlie local model biases better than the time-invariant local correction term used in the Delta Method, and indeed to an extent that results in more accurate estimates of past model biases. Similarly, Quantile Mapping assumes that the distributional correction of climate quantiles – whilst, again, not perfectly eliminating biases in present-day simulations – ultimately represents a better strategy for minimising past bias than the rigid local correction of the Delta Method.

Another important commonality between the methods is that they are calibrated only using present-day simulated and observed data. All three are based on the concept of establishing a relationship between present-day simulated and observed data, and then extrapolating that relationship in order to estimate past biases. The specific aspect that is assumed to be invariant over time is the present-day local bias in the case of the Delta Method, the regression model linking present-day simulated and observed data in the case of GAMs, and the present-day distributional correction in the case of Quantile Mapping.

### 2.3 Method evaluation

Empirical palaeoclimate reconstructions of climatological normals allow us to assess the performance of different bias correction methods in removing biases in past simulated data. In the following, we define the local differences between empirical reconstructions and bias-corrected simulations for the different climate variables

and bias correction method considered, and develop a spatially aggregated measure to assess the global performance of each method.

We denote by $V_{\text{emp}}(x,t)$ the empirically reconstructed value of a climate variable $V$ (representing terrestrial mean temperature, marine mean annual temperature, temperature of the coldest month, temperature of the warmest month, or annual precipitation) at a time $t$ in a location $x$. For a bias correction method $M$ (representing the Delta Method, GAM-based statistical bias correction, Quantile Mapping), we denote by $V_{\text{sim}}^{M}(x,t)$ the simulated value of the climate variable $V$ at the time $t$ in the location $x$ that was processed using the method $M$. The remaining local bias, $B$, between the empirically reconstructed and the bias-corrected simulation data at time $t$ in the location $x$ is then given by

$$B_V^M(x,t) = \begin{cases} V_{\text{sim}}^M(x,t) - V_{\text{emp}}(x,t) & \text{if } V \text{ is a temperature variable} \\ \dfrac{V_{\text{sim}}^M(x,t) - V_{\text{emp}}(x,t)}{V_{\text{emp}}(x,t)} & \text{if } V \text{ is annual precipitation} \end{cases} \tag{6}$$

Thus, we used the absolute difference between empirical and simulated data for temperature variables, and the relative difference in the case of precipitation. We denote by $x_1^{(t,V)}, x_2^{(t,V)}, \ldots$ the geographical locations of the available empirical records at time $t$ for the climate variable $V$. In section 3, we provide the complete distributions of the local biases that were derived for each climate variable, point in time, and bias correction method. In addition, as a summary statistic of these distributions and a spatially aggregated measure for evaluating the performance of each bias correction methods, we used the median of the available local absolute biases $\{|B_V^M(x_i^{(t,V)},t)|\}_{i=1,2,\ldots}$. The median is weighted by grid cell area for the present, and by the local inverse standard errors of the empirical data for the past. We rescaled the latter proportionally so that their sum equals 1, and denote the result by $\{\omega_{\text{emp}}(x_i^{(t,V)},t)\}_{i=1,2,\ldots}$ (i.e. $\sum_i \omega_{\text{emp}}(x_i^{(t,V)},t) = 1$). Formally, the median absolute bias, *MAB*, for the variable $V$ and the bias correction method $M$ at time $t$ is given by

$$\begin{aligned} MAB_V^M(t) &= \text{weighted median}\left(\{|B_V^M(x_i^{(t,V)},t)|\}_{i=1,2,\ldots}\right) \\ &= |B_V^M(x_k^{(t,V)},t)|, \text{ where the median index } k \text{ satisfies} \\ &\sum_{\substack{|B_V^M(x_i^{(t,V)},t)|<\ldots \\ |B_V^M(x_k^{(t,V)},t)|}} \omega_{\text{emp}}(x_i^{(t,V)},t) \le \frac{1}{2} \text{ and } \sum_{\substack{|B_V^M(x_i^{(t,V)},t)|>\ldots \\ |B_V^M(x_k^{(t,V)},t)|}} \omega_{\text{emp}}(x_i^{(t,V)},t) \le \frac{1}{2}. \end{aligned} \tag{7}$$

Thus, the median absolute bias is a measure of the average difference between empirical and the bias-corrected simulated data. We considered a bias correction method to overall improve the raw simulation outputs if the

associated median absolute bias is smaller than the median absolute difference between raw simulations and empirical data. The local biases $B$ in Eq. (6) and the *MAB* only permit to assess the performace of the three methods in bias-corrrecting quasi-quilibrated climatological normals of the variables considered here, not of climatic signals that are not captured by the underlying data, such as short-term climatic variability.

We tested whether the median absolute biases associated with any two bias correction methods, a certain climate variable and point in time, were statistically significantly different under the given uncertainty in the empirical reconstructions, using the following approach. For each climate variable and point in time, we generated $10^4$ Monte Carlo realisations of empirical past climatic values in the locations where reconstructions are available by applying a normally-distributed noise term, with mean zero and standard deviation equal to

the error of the local empirical reconstruction, to the value provided by the empirical reconstruction. Next, we calculated the local absolute biases between these empirical past climatic values, and the relevant simulated values obtained after applying the different bias correction methods. For each of these $10^4$ sets of local absolute biases between empirical and simulated data, we used a one-sided Wilcoxon rank sum test to assess whether the median of the absolute biases associated with one bias correction method was significantly smaller than that

associated with a different bias correction method (at a 5% significance level). We then determined the number of iterations, out of the total $10^4$ Monte Carlo realisations, in which this was the case. If, for a given climate variable and point in time, a bias correction method was found to perform significantly better than another one in more than half of the realisations, we report this result in section 3.

     Debiased simulated data should ideally not contain any systematic bias, in that the median bias, *MB*, given

by

$$MB_V^M(t) = \text{weighted median}\big(\{B_V^M(x_i^{(t,V)},t)\}_{i=1,2,\dots}\big), \tag{8}$$

where the weighted median is calculated analogously as in Eq. (7), should not differ substantially from zero. In addition to the median absolute bias (Eq. (7)), we also examined how the different methods affect the associated median bias (Eq. (8)).

In some applications, the climate change signal, i.e. the difference between past and present climatic states, may be more relevant than the climate at a fixed point in time. The difference between the empirical and the simulated climate change signal, *CCB*, of a climate variable $V$ that was bias-corrected using method $M$ at a

location $x$ and between the present and time $t$ in the past is calculated as

$$CCB_V^M(x_i^{(t,V)},t) = \begin{cases} \left(V_{\text{sim}}^M(x,t) - V_{\text{sim}}^M(x,0)\right) - \left(V_{\text{emp}}(x,t) - V_{\text{emp}}(x,0)\right) \\ \qquad\qquad\qquad\qquad\qquad \text{if } V \text{ is a temperature variable} \\ \dfrac{V_{\text{sim}}^M(x,t) - V_{\text{sim}}^M(x,0)}{V_{\text{sim}}^M(x,0)} - \dfrac{V_{\text{emp}}^M(x,t) - V_{\text{emp}}(x,0)}{V_{\text{emp}}(x,0)} \\ \qquad\qquad\qquad\qquad\qquad \text{if } V \text{ is annual precipitation,} \end{cases} \qquad (9)$$

and the median absolute bias associated with the climate change signal, *CCMAB*, is given by

$$CCMAB_V^M(t) = \text{weighted median}\left(\{|CCB_V^M(x_i^{t,V},t)|\}_{i=1,2,\dots}\right), \qquad (10)$$

where the weighted median is calculated analogously as in Eq. (6). We also compared the performance of the three bias correction methods in terms of this quantity. We did not determine the median absolute bias for the climate change signal between different points in the past, due to the much smaller number of empirical records that are available from the same location across the past, and due to the increased uncertainty of the local empirical climate change signals, which are given by the sum of the uncertainties of the local reconstructions of
the relevant points in time.

## 3   Results

Fig. 1a–e compare empirically reconstructed and bias-corrected simulated data for the five climate variables considered. They show that the biases that remain after applying a bias correction method are not uniformly distributed across the range of simulated values. In a number of cases, very low temperatures in several bias-
corrected simulations tend to be lower than empirically reconstructed values, while very high temperatures in the simulated data tend to be higher than what empirical reconstructions suggest (e.g. Mid-Holocene and Last Interglacial mean annual marine temperature, and Mid-Holocene and LGM temperature of the warmest month). For some bias correction methods, an analogous patterns can be observed in the case of precipitation.

   All bias correction methods reduce the median absolute bias (*MAB* in Eq. (6)) of present-day simulated
data for all climate variables, as would be expected (Maraun and Widmann, 2018) (Fig. 2). By construction, the Delta Method completely eliminates all differences between present-day simulated and observed data. The Delta Method also provides the strongest reduction in the median absolute bias (*MAB* in Eq. (6)) for all variables and

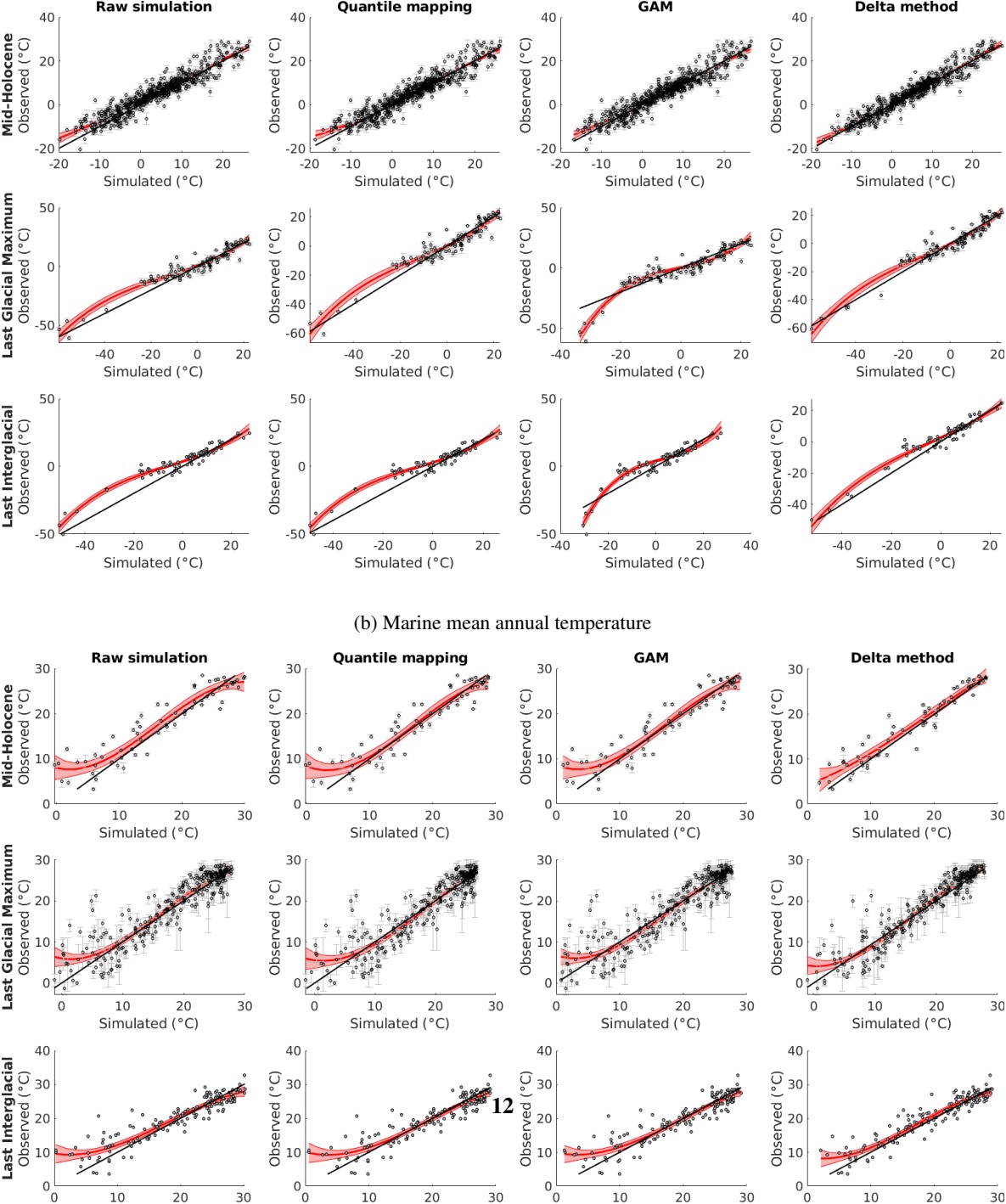

(a) Terrestrial mean annual temperature

(b) Marine mean annual temperature

## (c) Mean temperature of the warmest month

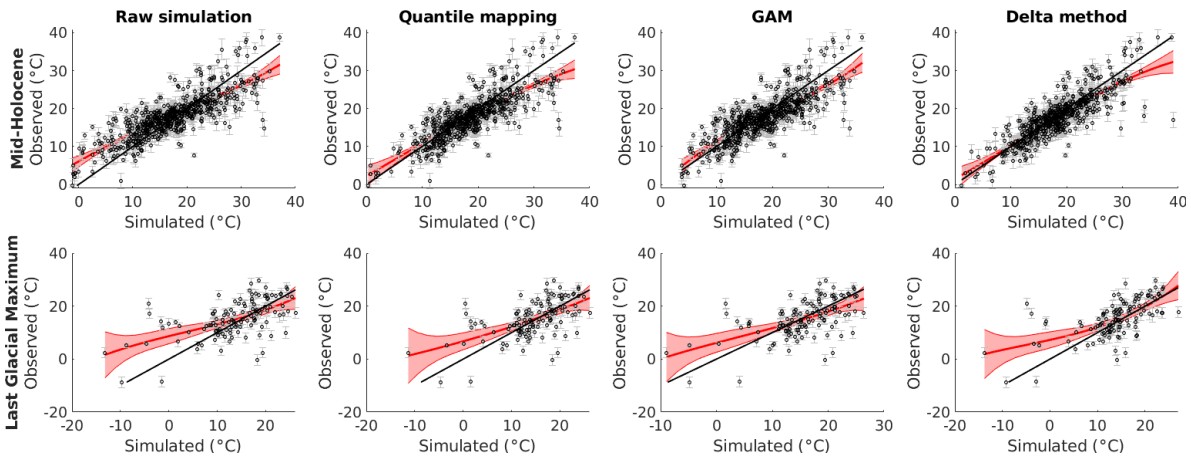

## (d) Mean temperature of the coldest month

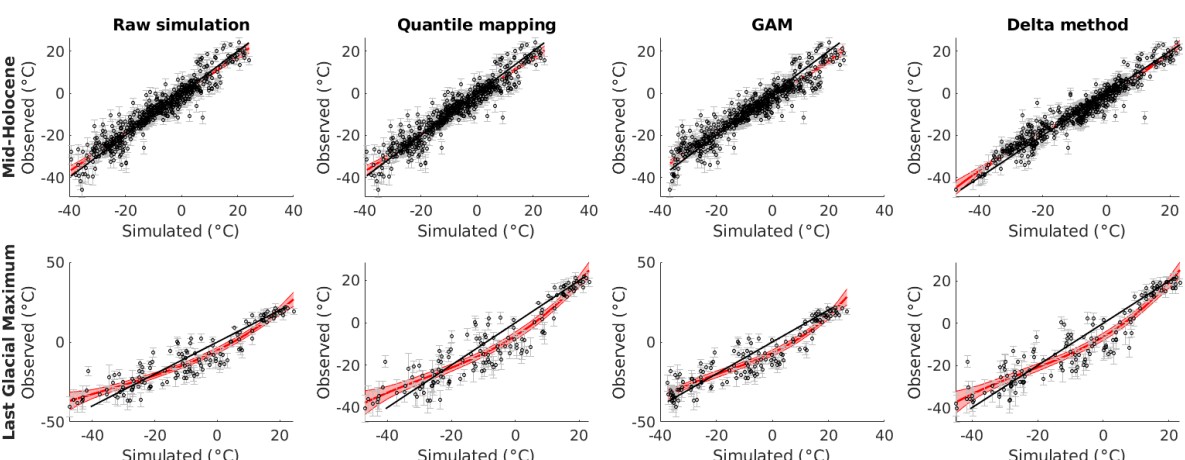

## (e) Annual precipitation

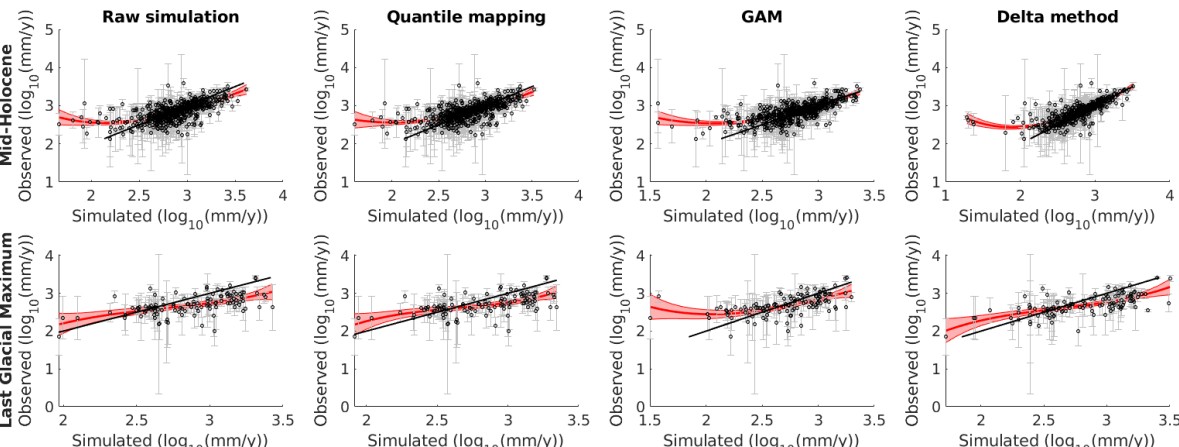

**Figure 1.** Comparison of bias-corrected simulated and empirically reconstructed climate variables. Black lines show 1:1 relationships. Red lines and shades show 5th degree polynomial regression and 95% confidence intervals, respectively.

points in time, with the expection of temperature of the coldest month at the Mid-Holocene and precipitation at the LGM (Fig. 2). The comparatively good performance of the Delta Method is reflected in strong correlations between present-day and past model biases, which the Delta Method assumes to be similar (Fig. A1). The GAM method and Quantile Mapping also generally lead to a reduction in bias, though overall not quite as strongly as

5  the Delta Method. In a few cases, the original bias is actually increased after applying a correction method (Fig. 2).

These above trends in the performances of the different bias correction methods in terms of the median absolute bias are not always statistically significant. The median absolute bias associated with the Delta Method was significantly smaller ($p < 0.05$) than that associated with Quantile Mapping and the GAM method for Mid-

10  Holocene terrestrial mean annual temperature (in 96% and 83% of Monte Carlo realisations (see section 2.3) when compared against Quantile Mapping and the GAM method, respectively), marine mean annual temperature (in 93% and 89% of realisations, respectively), terrestrial mean temperature of the warmest month (in 92% and 100% of realisations, respectively), and precipitation (in 100% and 100% of realisations, respectively). The Delta Method also performed significantly better than the GAM method for Mid-Holocene terrestrial mean tem-

perature of the coldest month (86% of realisations), and significantly better than Quantile Mapping for LGM marine mean annual temperature (65% of realisations). The GAM method performed significantly better than Quantile Mapping for LGM precipitation (100% of realisations). By design, the Delta Method has a significantly lower median absolute bias (namely zero) than both other methods for all variables at present day.

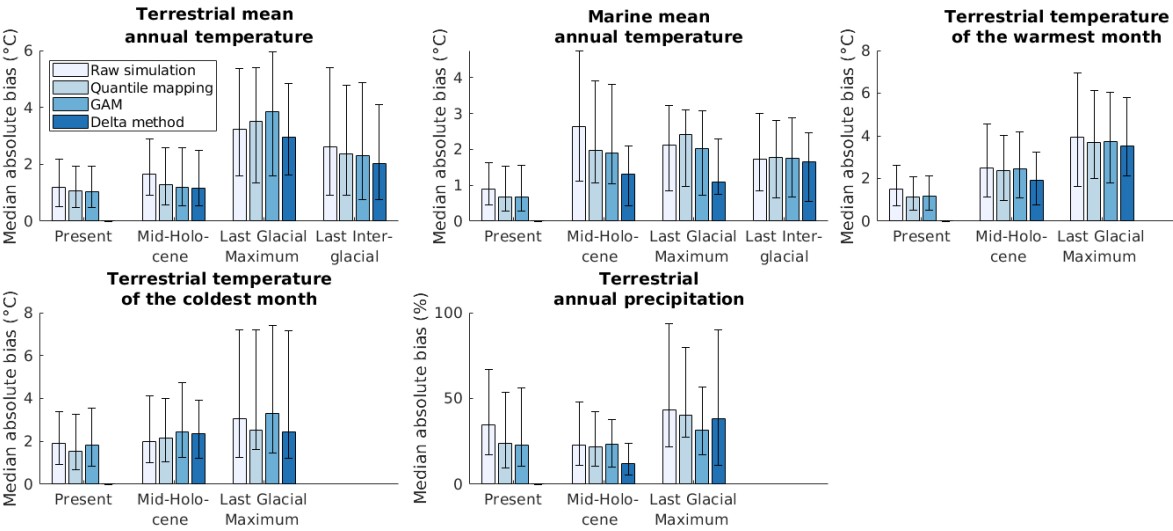

**Figure 2.** Median absolute biases (*MAB*, Eq. (7)) of the raw and bias-corrected climate simulation data. Error bars represent 25% and 75% weighted quantiles of the local absolute biases available for the given climatic variable and point in time.

5    Across time periods, raw simulations tended to underestimate terrestrial and marine mean annual temperature and terrestrial temperature of the warmest month, and overestimated annual precipitation (Fig. A2). These trends are less present in the bias-corrected data: methods consistently reduced the absolute value of the median bias (*MB* in Eq. (8)) of the raw simulations, except in the case of terrestrial temperature of the coldest month.

The differences between bias correction methods in terms of improving the climate change signal (*CCMAB* in Eq. (10)) are negligible in all scenarios except for marine mean annual temperature during the Last Glacial 10   Maximum, where the GAM method performs slightly better than other methods (Fig. A3).

The performance of the different methods is not uniform across space nor time. Fig. 3 illustrates this heterogeneity for the Delta Method. For example, the Delta Method significantly reduces the original bias of modelled

precipitation in Eastern North America in the Mid-Holocene, but hardly improved the raw simulations in the Sahara, whereas the opposite pattern can be observed at the LGM.

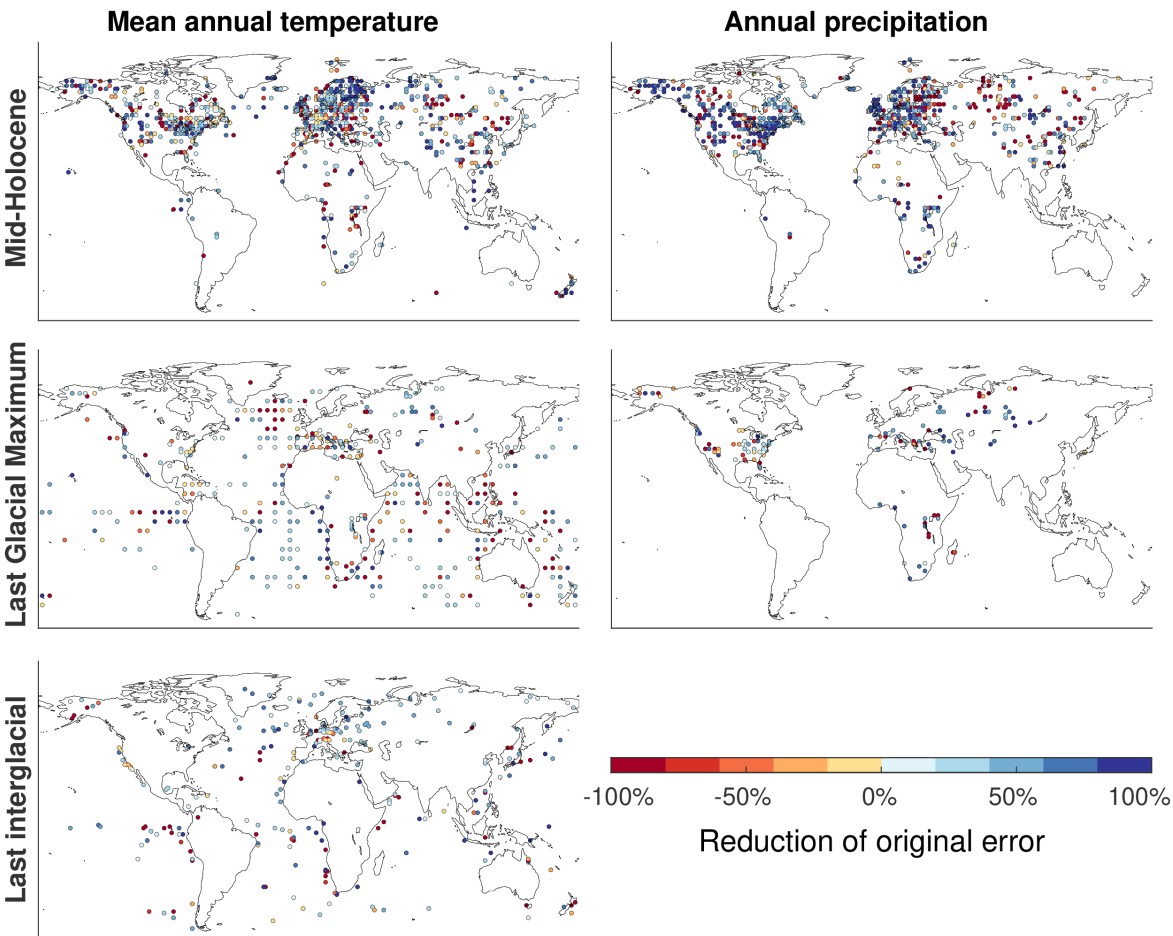

**Figure 3.** Reduction of the original model bias by the Delta Method for terrestrial and marine mean annual temperature and terrestrial annual precipitation. The lower end of the colour scale was capped at -100% (i.e. a doubling of the original bias).

The performances of the methods relative to each other also vary substantially across both space and time. For example, whilst globally the Delta Method has a slight overall edge over the GAM approach (Fig. 2), the

comparison of the two methods in Fig. 4 shows that even within small geographical regions neither method performs consistently better than the other. Moreover, a better performance of one method in a certain location at some point in time generally does not guarantee the same result at a different time. For instance, the Delta Method overall reduced the original bias of modelled precipitation more than the GAM approach in Eastern North America during the Mid-Holocene, but less during the LGM (Fig. 4).

## 4 Discussion

Whilst, overall, the Delta Method performs slightly better at debiasing temperature and precipitation compared to the GAM-based method and Quantile Mapping for the empirical data considered here, we note that this method is only appropriate for a given land conformation. Thus, it is only suitable for the Late Quaternary, and even for this period, changes in sea levels are problematic as they expose areas for which there is no bias information. GAMs should, in theory, obviate this problem by quantifying bias-related processes as statistical relationships; however, whilst this approach might be the only option for the deeper past, our results point to the fact that estimating such processes in such a way is challenging, as demonstrated by its overall inferior performance to the Delta Method. A possible limitation of GAMs as currently applied is that they assume additivity between predictor variables. By fitting interactions, it would be possible to allow for more complex processes, but the computational complexitiy of interactions with such large datasets is non-trivial.

A major limitation of current approaches for bias-correcting climate model data is that they all assume bias patterns in present-day climate to be fully representative of the past (see section 2.2.4). With the progressive increase in the number of empirical records of past climatic conditions, it may be possible to soon move from a situation where past reconstructions are use to verify bias correction methods (as we did in this manuscript) to using those data to actively calibrate bias correction methods. Despite large uncertainties, and patterns that are not fully consistent across time, Fig. 5 suggests an intriguing relationship between the variation of local model biases across time on the one hand, and simulated climate change signals on the other hand. Such a relationship could, in principle, be used to refine the Delta Method by accounting for the change of local model biases over

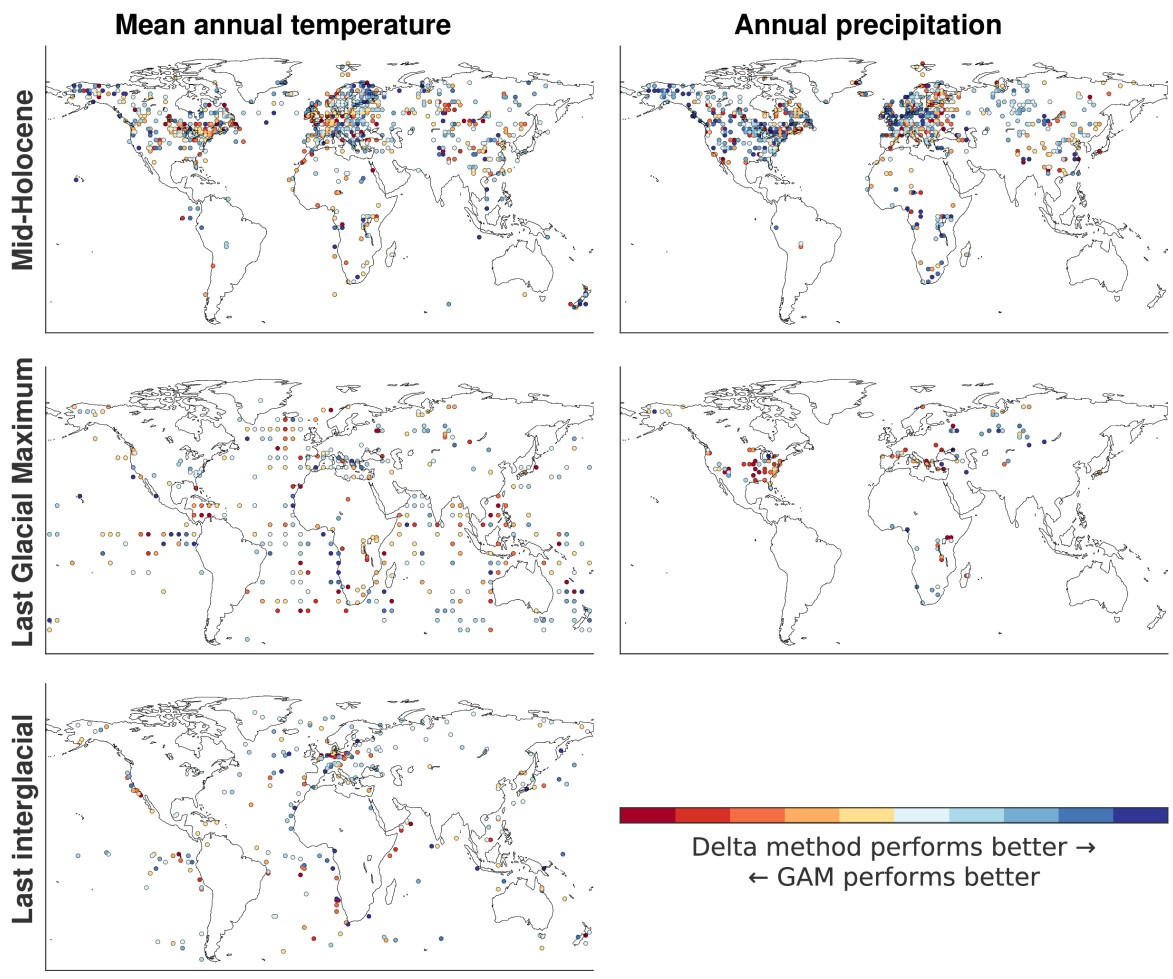

**Figure 4.** Relative performances of the Delta Method and the GAM approach in terms of debiasing simulated mean annual temperature (left column) and annual precipitation (right column). The colour spectrum represents the interval [0,1], and marker colours are calculated as the ratio of the absolute value of the local bias (Eq. (6)) of the GAM-based approach divided by the sum of the absolute local biases of both methods.

time. For example, instead of Eq. (1), we would have

$$T_{\text{sim}}^{DM+}(x,t) := T_{\text{sim}}^{raw}(x,t) + \underbrace{\left(T_{\text{emp}}(x,0) - T_{\text{sim}}^{raw}(x,0)\right)}_{\substack{\text{standard time-invariant Delta} \\ \text{Method bias correction term}}} + \underbrace{f\left(\underbrace{T_{\text{sim}}^{raw}(x,t) - T_{\text{sim}}^{raw}(x,0)}_{\substack{\text{simulated climate} \\ \text{change signal}}}, \underbrace{\ldots}_{\substack{\text{additional} \\ \text{predictor} \\ \text{variables}}}\right)}_{\text{time-variable correction term}}, \tag{11}$$

where $f$ represents a non-linear regression model satisfying $f|_{t=0} = 0$. A robust statistical model $f$ will require not only additional data from currently underrepresented geographical areas (specifically the southern hemisphere), but also the curation of empirical reconstructions, as successfully done for the last millenium (Hakim et al., 2016; Tardif et al., 2018).

Such an approach would tie in with data assimilation methods, which also use empirical climate proxy records to improve climate simulations. These methods have been used to estimate global climate variables at times at which the quantity and spatial coverage of available empirical records is high enough to allow a robust calibration of the relevant computational methods. As a result, they have focussed either on single points in the past, such as the Mid-Holocene (Mairesse et al., 2013) or Last Glacial Maximum (Kurahashi-Nakamura

et al., 2017), or on time intervals across which suitable empirical data are available, namely the last millennium period (Tardif et al., 2019; Goosse, 2017). In contrast to the aforementioned approaches, based on Fig. 5 we suggest that it may be possible to use empirical reconstructions even from only a small set of points in time (e.g. the present, Mid-Holocene, LGM and Last Interglacial Period) to parameterise a statistical model of the temporal variation of local biases that could be used to improve simulated data at any time point in the Late

Pleistocene-Holocene period.

## 5    Conclusions

Our comparison of global debiased palaeosimulation data and empirical reconstructions suggests that, despite its conceptual simplicity, the Delta Method is good starting point for bias correction of simulated Late Quaternary climate data at a global scale, providing slightly stronger bias reductions compared to GAMs and Quantile

Mapping. However, given the lack of statistical significance of the superior performance in some cases, and the considerable variability in the effectiveness of the three methods across different locations and points in time, we echo earlier propositions that studies focussing on specific regions require case-by-case assessments of which bias correction method is most suitable for improving palaeoclimate simulations (Maraun et al., 2017). We also reiterate that our results may be different for palaeoclimate simulations other than the ones used here. Finally, it

is important to bear in mind that bias correction methods are unable to substantially correct a fundamentally poor climate model, e.g. with strong circulation biases, which such methods are not capable of removing (Maraun

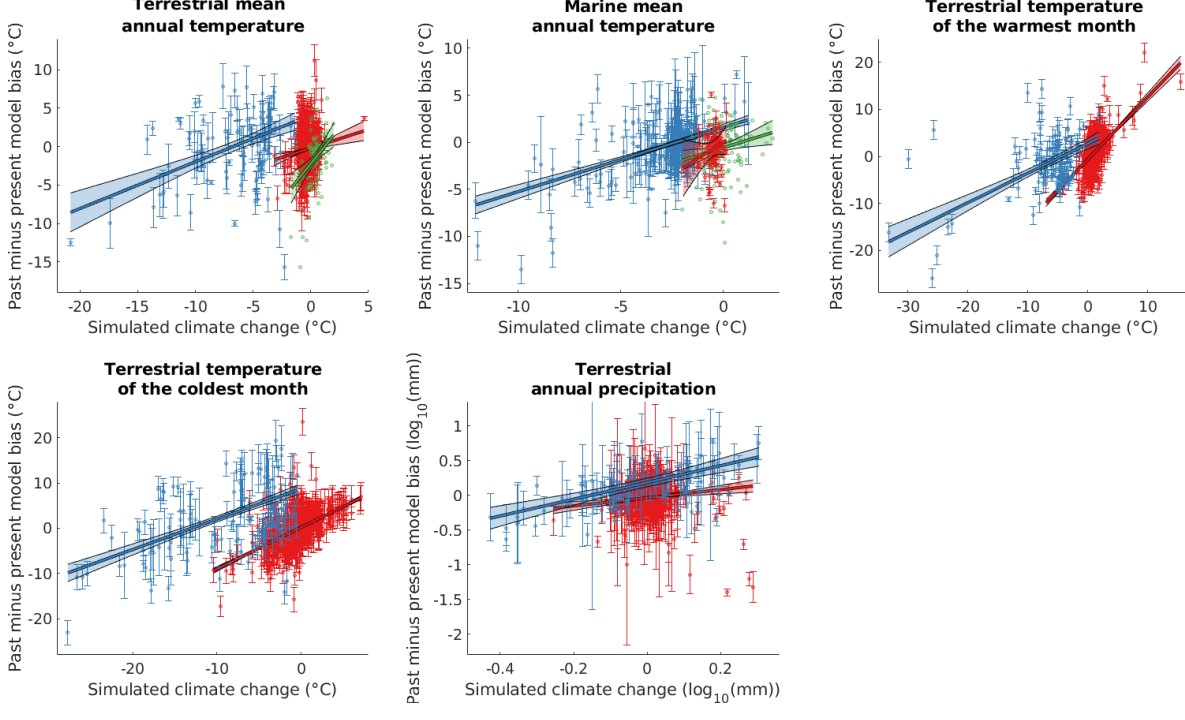

**Figure 5.** Differences between local past and present model bias (at locations for which empirical reconstructions are available) against the local simulated climate change signal (i.e. the difference between past and present simulated value) of the variable of interest. Red, blue and green markers represent data from the Mid-Holocene, the LGM and the Last Interglacial Period, respectively. Error bars represent standard errors of the empirical reconstructions. Lines and shades show robust linear regressions and 95% confidence intervals, respectively. Whilst weak, the relationships suggest that it may be possible to model some of the variability of local model biases over time, using only available simulation data. Such an approach could potentially significantly enhance the Delta Method, which currently operates on the simplifying assumption that this variability is negligible.

et al., 2017). Seeking to improve the representation of climate dynamics in simulation models therefore remains a priority alongside the development of bias correction methods.

A key limitation of all three methods considered here is their assumption that present-day patterns between simulated and observed climate can be extrapolated to estimate model biases in the past. High uncertainties, and

the spatial and temporal sparseness associated with currently available empirical palaeoclimate datasets will likely impede a robust assimilation of these data into bias correction methods at this stage; however, our data indicate the increasing quantity and quality of global proxy records could soon make it possible to use empirical reconstructions in the development of improved methods that effectively account for the variation of local model

5    biases through time.

*Code and data availability.*  Code and datasets used in this analysis will be made publicly available on the Open Science Framework repository upon acceptance of the manuscript.

*Author contributions.*  All authors conceived the study. R.B. conducted the analysis and wrote the manuscript. All authors interpreted the results and revised the manuscript.

10   *Competing interests.*  The authors declare no competing interests.

*Acknowledgements.*  The authors are grateful to Paul J. Valdes and Joy S. Singarayer for providing the climate simulation data used in this study, and to three anonymous reviewers for their helpful comments. R.B., M.K. and A.M. were supported by the ERC Consolidator Grant 647787 ("LocalAdaptation").

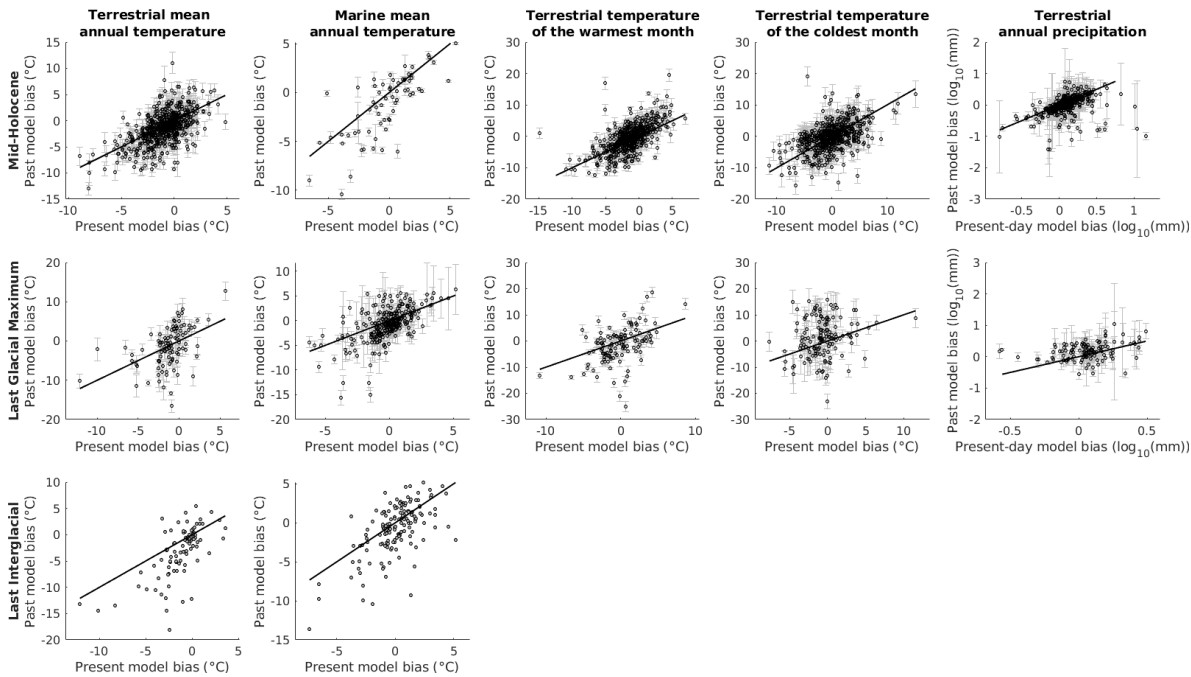

**Figure A1.** Comparison of present-day and past model biases (which the Delta Method assumes to be similar) from locations where empirical reconstructions are available. Lines represent 1:1 relationships.

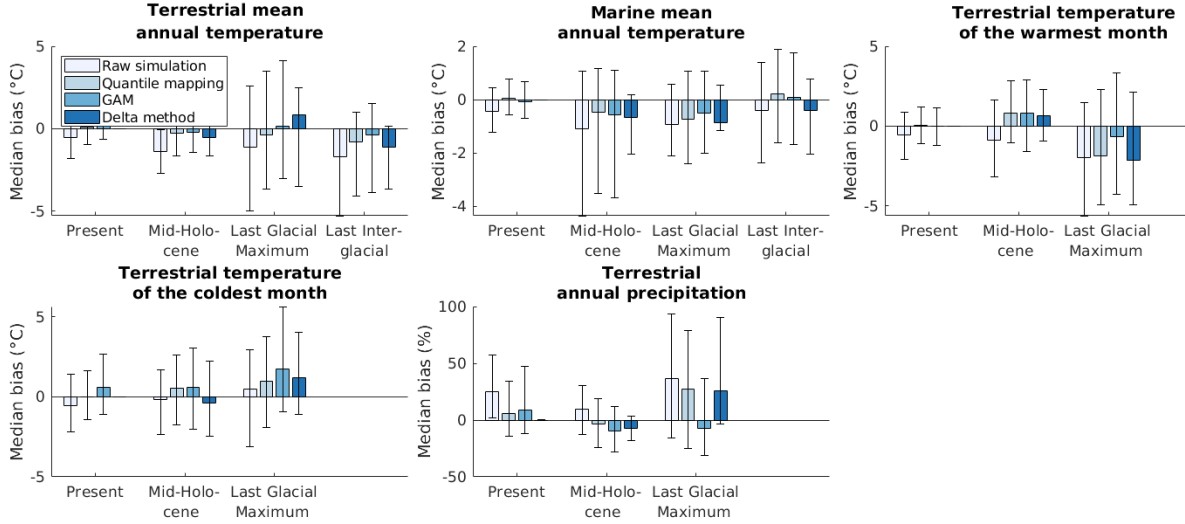

**Figure A2.** Median biases of the raw and bias-corrected climate simulation data. Error bars represent 25% and 75% weighted quantiles of the local biases available for the given climatic variable and point in time.

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

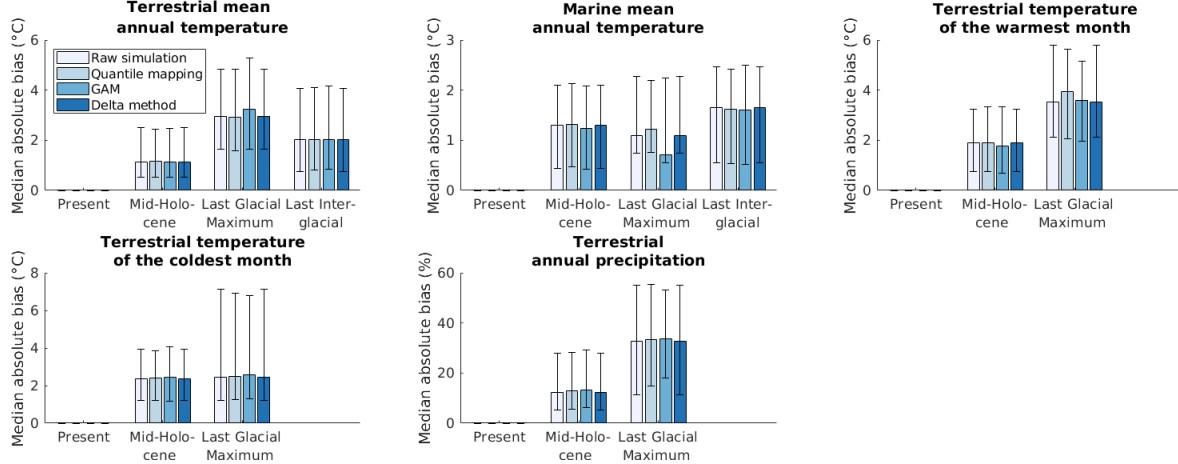

**Figure A3.** Median absolute biases of the climate change signal (*CCMAB*, Eq. (10)). Error bars represent 25% and 75% weighted quantiles of the local absolute climate change biases available for the given climatic variable and point in time.

Eriksson, A., Betti, L., Friend, A. D., Lycett, S. J., Singarayer, J. S., von Cramon-Taubadel, N., Valdes, P. J., Balloux, F., and Manica, A.: Late Pleistocene climate change and the global expansion of anatomically modern humans, Proceedings of the National Academy of Sciences, 109, 16 089–16 094, 2012.

Goosse, H.: Reconstructed and simulated temperature asymmetry between continents in both hemispheres over the last
5    centuries, Climate Dynamics, 48, 1483–1501, 2017.

Hakim, G. J., Emile-Geay, J., Steig, E. J., Noone, D., Anderson, D. M., Tardif, R., Steiger, N., and Perkins, W. A.: The last millennium climate reanalysis project: Framework and first results, Journal of Geophysical Research: Atmospheres, 121, 6745–6764, 2016.

Hessler, I., Harrison, S., Kucera, M., Waelbroeck, C., Chen, M.-T., Anderson, C., De Vernal, A., Fréchette, B., Cloke-Hayes,
10    A., Leduc, G., et al.: Implication of methodological uncertainties for mid-Holocene sea surface temperature reconstructions, Climate of the Past, 10, 2237–2252, 2014.

Ho, C. K., Stephenson, D. B., Collins, M., Ferro, C. A. T., and Brown, S. J.: Calibration Strategies: A Source of Additional Uncertainty in Climate Change Projections, Bulletin of the American Meteorological Society, 93, 21–26, https://doi.org/10.1175/2011BAMS3110.1, https://journals.ametsoc.org/doi/abs/10.1175/2011BAMS3110.1, 2011.

15  Hudson, D. and Jones, R.: Regional climate model simulations of present-day and future climates of southern Africa, Hadley Centre for Climate Prediction and Research, 2002.

Kottek, M., Grieser, J., Beck, C., Rudolf, B., and Rubel, F.: World map of the Köppen-Geiger climate classification updated, Meteorologische Zeitschrift, 15, 259–263, 2006.

Krapp, M., Beyer, R., Edmundsson, S. L., Valdes, P. J., and Manica, A.: A comprehensive climate history of the last 800 thousand years, EarthArXiv, https://doi.org/10.31223/osf.io/d5hfx, 2019.

Kurahashi-Nakamura, T., Paul, A., and Losch, M.: Dynamical reconstruction of the global ocean state during the Last Glacial Maximum, Paleoceanography, 32, 326–350, 2017.

Latombe, G., Burke, A., Vrac, M., Levavasseur, G., Dumas, C., Kageyama, M., and Ramstein, G.: Comparison of spatial downscaling methods of general circulation model results to study climate variability during the Last Glacial Maximum, Geoscientific Model Development, 11, 2563–2579, 2018.

Leonardi, M., Boschin, F., Giampoudakis, K., Beyer, R. M., Krapp, M., Bendrey, R., Sommer, R., Boscato, P., Manica, A., Nogues-Bravo, D., et al.: Late Quaternary horses in Eurasia in the face of climate and vegetation change, Science advances, 4, eaar5589, 2018.

Levavasseur, G., Vrac, M., Roche, D., Paillard, D., Martin, A., and Vandenberghe, J.: Present and LGM permafrost from climate simulations: contribution of statistical downscaling, Climate of the Past, 7, 1225–1246, 2011.

Lorenz, D. J., Nieto-Lugilde, D., Blois, J. L., Fitzpatrick, M. C., and Williams, J. W.: Downscaled and debiased climate simulations for North America from 21,000 years ago to 2100AD, Scientific data, 3, 160 048, 2016.

Mairesse, A., Goosse, H., Mathiot, P., Wanner, H., and Dubinkina, S.: Investigating the consistency between proxy-based reconstructions and climate models using data assimilation: a mid-Holocene case study, Climate of the Past, 9, 2741–2757, 2013.

Maraun, D. and Widmann, M.: Statistical downscaling and bias correction for climate research, Cambridge University Press, 2018.

Maraun, D., Shepherd, T. G., Widmann, M., Zappa, G., Walton, D., Gutiérrez, J. M., Hagemann, S., Richter, I., Soares, P. M., Hall, A., et al.: Towards process-informed bias correction of climate change simulations, Nature Climate Change, 7, 764, 2017.

New, M., Lister, D., Hulme, M., and Makin, I.: A high-resolution data set of surface climate over global land areas, Climate research, 21, 1–25, 2002.

Rangel, T. F., Edwards, N. R., Holden, P. B., Diniz-Filho, J. A. F., Gosling, W. D., Coelho, M. T. P., Cassemiro, F. A., Rahbek, C., and Colwell, R. K.: Modeling the ecology and evolution of biodiversity: Biogeographical cradles, museums, and graves, Science, 361, eaar5452, 2018.

Reynolds, R. W., Rayner, N. A., Smith, T. M., Stokes, D. C., and Wang, W.: An improved in situ and satellite SST analysis for climate, Journal of climate, 15, 1609–1625, 2002.

Solomon, S., Qin, D., Manning, M., Chen, Z., Marquis, M., Avery, K., Tignor, M., and Miller, H., eds.: IPCC: Climate Change 2007: The Physical Science Basis. Contribution of Working Group I to the Fourth Assessment Report of the Intergovernmental Panel on Climate Change, Cambridge University Press, 2007.

Tardif, R., Hakim, G. J., Perkins, W. A., Horlick, K. A., Erb, M. P., Emile-Geay, J., Anderson, D. M., Steig, E. J., and
Noone, D.: Last Millennium Reanalysis with an expanded proxy database and seasonal proxy modeling, Climate of the Past Discussions, pp. 1–37, 2018.

Tardif, R., Hakim, G. J., Perkins, W. A., Horlick, K. A., Erb, M. P., Emile-Geay, J., Anderson, D. M., Steig, E. J., and Noone, D.: Last Millennium Reanalysis with an expanded proxy database and seasonal proxy modeling., Climate of the Past, 15, 2019.

Timmermann, A. and Friedrich, T.: Late Pleistocene climate drivers of early human migration, Nature, 538, 92, 2016.

Turney, C. S. and Jones, R. T.: Does the Agulhas Current amplify global temperatures during super-interglacials?, Journal of Quaternary Science, 25, 839–843, 2010.

Valdes, P. J., Armstrong, E., Badger, M. P., Bradshaw, C. D., Bragg, F., Davies-Barnard, T., Day, J. J., Farnsworth, A., Hopcroft, P. O., Kennedy, A. T., et al.: The BRIDGE HadCM3 family of climate models: HadCM3@ Bristol v1. 0,
Geoscientific Model Development, 10, 3715–3743, 2017.

Vrac, M., Marbaix, P., Paillard, D., and Naveau, P.: Non-linear statistical downscaling of present and LGM precipitation and temperatures over Europe, Climate of the Past, 3, 669–682, 2007.

Waelbroeck, C., Paul, A., Kucera, M., Rosell-Melé, A., Weinelt, M., Schneider, R., Mix, A. C., Abelmann, A., Armand, L., Bard, E., et al.: Constraints on the magnitude and patterns of ocean cooling at the Last Glacial Maximum, Nature
Geoscience, 2, 127, 2009.

Woillez, M.-N., Levavasseur, G., Daniau, A.-L., Kageyama, M., Urrego, D., Sánchez-Goñi, M.-F., and Hanquiez, V.: Impact of precession on the climate, vegetation and fire activity in southern Africa during MIS4, Climate of the Past, 10, 1165–1182, 2014.

Wood, S. N.: Stable and efficient multiple smoothing parameter estimation for generalized additive models, Journal of the
American Statistical Association, 99, 673–686, 2004.

Zhu, D., Ciais, P., Chang, J., Krinner, G., Peng, S., Viovy, N., Peñuelas, J., and Zimov, S.: The large mean body size of mammalian herbivores explains the productivity paradox during the Last Glacial Maximum, Nature ecology & evolution, 2, 640, 2018.