# Peer review of "An empirical evaluation of bias correction methods for palaeoclimate simulations"

_Climate of the Past, 2019_

## Referee Comment (RC1) · Anonymous Referee #1 · 1 Mar 2019

The manuscript is focused on the evaluation of three statistical methods to correct the biases of the output of paleoclimate simulations. This evaluation is carried out by comparing the corrected simulated values of annual temperature and annual precipitation with the corresponding reconstructions based on proxy record for three periods in the past (Mid-Holocene, Last Glacial Maximum and Last Interglacial Period). The three statistical methods are the Delta Method, Quantile Mapping and Generalized Additive Models that include additional geographical predictors to correct the simulated climatologies. The main conclusion is that, in general, the Delta Method outperforms the other two.

My general impression of the manuscript is positive, but I have two main concerns that I think should be addressed in a revised version. In addition, I have a few more

suggestions to particular points in the manuscript that would benefit from a further clarification.

My evaluation is that the manuscript would need some (not drastic) revisions, but I would like to evaluate the revised version.

Main concerns

1) The evaluation criterion is essentially the difference between the corrected and reconstructed climatology. However, the Delta Method has been specifically constructed to eliminate this difference between simulated climatology and present-day climatology. Quantile Mapping pursues a more general correction, namely to correct the whole probability distribution of annual temperature (or precipitation). The GAM method is a statistical model that incorporates (in my understanding) simulated and observed grid-point climatologies as predictors and predictands , and additionally some other factors like distance to the ocean, etc. The GAM method is therefore also not specifically tailored to eliminate the bias. I wonder if the main result of the manuscript, namely the best performance of the Delta method, is not an artifact. The Delta Method is precisely tailored to maximise the evaluation criterion and thus , it is for me not surprising that it outperforms the other two methods. I am not sure which other, fairer, evaluation criterion could be introduced, but I think that this issue should be addressed or at least thoroughly discussed.

2) The difference between the corrected simulated climatologies and the reconstructed climatologies does not take into account the presumably large uncertainty in the reconstructions and in the corrected simulated climatologies (the former being presumably much larger?) . This needs to be incorporated in the evaluation of the three methods. If the inter-methodological differences are much smaller than the uncertainties in the estimated paleo-bias , it would be difficult to claim that one particular method is superior to other two. I think that the manuscript should include also these uncertainty estimations, or at least place the inter-methodological differences in the frame of the

reconstruction uncertainties.

3) The readability of the illustrations is poor. it is, for instance, very difficult to discern anything in Figure 2 and Figure 3. The lettering, axis labels, etc, in most figures is too small (e.g Figure 4)

Particular points

4) what is the original spatial resolution of the climate reconstructions ? were they regridded, and how?

5) The text refers sometimes to bias , other times to 'error', whereas in my understanding very often both terms carry the same meaning. This can be confusing for some readers. I would recommend to stick to one of those terms when possible.

6) The text also refers to the climate reconstructions as 'the observations', e-g. in equation 5. This can also be confusing. It would be clearer to use 'climate reconstructions' when referring to the proxy-reconstructed climatologies and 'observations' when referring to present-day climatologies.

7) The main conclusion is derived from the analysis of only one model. Perhaps I missed it but I think this a caveat that should be mentioned.
* * *

---

## Referee Comment (RC2) · Anonymous Referee #2 · 1 Jun 2019

Review of

'A systematic comparison of bias correction methods for palaeoclimatic simulations'

by R. Beyer, M. Krapp and A. Manica

**Recommendation: reject, invite resubmission of a revised version**

This manuscript presents a comparison of three bias correction (BC) methods applied to GCM simulations for the present, Mid-Holocene, and Last Glacial Maximum. The method applied are a delta change (DC) method, a generalized linear model (GLM), and quantile mapping (QM).

Exploring BC methods in the context of palaeoclimate simulations is a useful contribution to the research area, and I thus in principle support publication of studies related to this topic. However the use of BC in this manuscript is mechanistic, uncritical and superficial, and the overall approach, methods and specific setups are poorly explained, There is also a potential implementation error for QM. The lack of clarity on what has actually been done is so large that I cannot assess whether the results are in principle suitable for publication. I thus think that a revised manuscript should be sent for a full review again.

**Specific comments**

- 1) Page 1, lines 11-13, The DC method has its name because it is based on adding the simulated difference between two periods to observations. Although this is mathematically equivalent to subtracting the fitting period bias from the simulations (as shown in eqn.1) I think introducing the BC methods using the second definition rather than the one that is directly linked to the name is potentially confusing.
- 2) Page 1, lines 13 -15, Please use clean terminology. GAM is a statistical representation of links between 'variables' not between 'proxies for processes' and 'biases'. Define clearly what predictors and predictands are. From the current statement it is impossible to find out which variables are actually linked through GAM.
- 3) Page 1, lines 15-16, QM does not assume the shape of the distribution to be constant in time. If there is climate change the distribution obviously changes. Standard implementations assume that the bias for a given value is constant in time (but there are implementations without this assumption). Please remove wrong statement and include a correct explanation.
- 4) Page 1, lines 18-20. Please be more specific about the potential setups in the palaeoclimate context. Some empirical palaeoclimate reconstructions are local or have a high spatial resolution, which means they are smaller-scale than the climate model output (downscaling), whereas continental-scale empirical reconstructions have a lower resolution than the models (upscaling).

5) Page 1, There is a complete lack of critical discussion about the limitations of BC. It is obvious that a fundamentally poor model cannot be improved in a meaningful way by BC (see for instance Maraun and Widmam (2018), Maraun et al., 2017: Towards process-informed bias correction of climate change simulations. *Nature Climate Change*, **7**(11), 764-773). These limitations should be discussed in the introduction, in particular in the context of palaeoclimate simulations.

Moreover, the validation approach needs to be justified taking into account the potential problems with BC, and clear comments need to be made on whether the validation would identify such problems. It will turn out that it would not (see comments below), which should at least be stated as a limitation of the study.

- 6) Page 4, lines 18-19, The statement about the log-transform is correct, but overcomplicated. It is more helpful to just say that this is a multiplicative delta method, i.e. the simulated relative change is applied to the observations.
- 7) Page 4, lines 21-22, The sentence doesn't work out. The relationship is between climate model output and real-world climate variables, with additional time-invariant predictors such as topography or distance from the coast.
- 8) Page 5, eqn. 3, Clarify that some x\_i are time-dependent (i.e. those that represent climate model output), while others (topography, distance from coast) are not.
- 9) Page 5, line 13, does 'wind speed' include the direction?
- 10) Page 5, line 14-15, It is not clear what the predictor and predictand data are and how the fitting for the f\_i works. What are the individual realisations of T\_sim and x\_i for which the polynomials are fitted? Are these timesteps? But if so, if I understand correctly, there are only three, namely the mean temperatures for the present, Mid-Holocene, and Last Glacial Maximum. What is the spatial resolution? Are the simulated temperatures averaged over the continental areas represented in the proxy-based reconstructions? Or are the realisations in space (if so, is this one value for each continent?), or space and time?
- 11) Page 5, line 17-22, It is not clear what the distributions are. Are they annual values of continental means?
- 12) Page 6, The evaluation method needs more justification. For instance it would be a logical first step to validate the three BC methods on instrumental data, using cross-validation, and focusing on aspect that are important in the palaeoclimatic context, i.e. long-term variability. The argument is probably that the key aspect is the representation of changes on multi-millennial timescales. The evaluation section should start with stating the objectives of the evaluations, followed by a justification of why the chosen evaluation method addresses these objectives.

Please keep in mind that BC methods reduce bias by construction, even for completely wrong models (see e.g. Maraun et al, 2017). A reduction in the bias of the mean (DC, GAM), will reduce also the biases for the distribution quantiles, while BC corrects these directly. For strongly biased climate models the reduction of the biases in the distributions will necessarily lead to a reduction in MAE. Why is the MAE chosen as the evaluation measure? In the paleoclimate context it is also very relevant to compare the

climate change signals in the raw and the BC-corrected simulations, and in the proxybased reconstructions. Please add statements and if suitable figures on this.

- 13) Page 5, line 7, 'standard errors' of what? It is said later that it is the error of the reconstructions, but it needs to be said the first time this is mentioned.
- 14) Page 6, eqn. 6, The notation is very unclear. It is also not clear what 'grid cell' refers to. Earlier it was mentioned that continental means are used. This problem is related to the lack of clarity about predictand and predictor data mentioned in previous comments.
- 15) Page 7, figure 1. It seems not plausible that QM leads to substantially larger MAEs than for the raw simulations, with values up to 10 K. Surprisingly this is not even discussed. There might be a problem with the implementation. If the implementation is correct, please give a detailed explanation how this is possible. If I understand correctly the BC-corrected distribution of the present simulation is identical to the distribution of the instrumental observations. This means that the instrumental observations have also a very high MAE for the present. How can this be the case? If suitable, please add information about how the instrumental data, which are the training data for all BC methods, perform in this evaluation framework.

When addressing this please state explicitly what is compared with what for calculating the MAE; the information that is currently given is incomplete.

16) Page 11, figure 4. If I understand correctly, this figure shows that the simulated climate change signal is different from the reconstructed climate change signal. If this is correct, please include this straightforward interpretation.

---

## Editor Comment (EC1) · Steven Phipps (Editor) · 3 Jun 2019

Dear authors,

Two reviewers have now posted comments on your manuscript. Both have raised substantive issues. Although I note that Referee #2 has recommended rejection, I am willing to consider a revised manuscript. However, this would be dependent upon you comprehensively revising the manuscript in line with the comments that you have received.

Therefore, please ensure that you respond carefully and thoroughly to each review comment.

Steven Phipps, Handling Editor

---

## Author Comment (AC1) · 17 Oct 2019

**Reviewer 1**

1) The evaluation criterion is essentially the difference between the corrected and reconstructed climatology. However, the Delta Method has been specifically constructed to eliminate this difference between simulated climatology and present-day climatology. Quantile Mapping pursues a more general correction, namely to correct the whole probability distribution of annual temperature (or precipitation). The GAM method is a statistical model that incorporates (in my understanding) simulated and observed gridpoint climatologies as predictors and predictands , and additionally some other factors like distance to the ocean, etc. The GAM method is therefore also not specifically tailored to eliminate the bias. I wonder if the main result of the manuscript, namely the best performance of the Delta method, is not an artifact. The Delta Method is precisely tailored to maximise the evaluation criterion and thus , it is for me not surprising that it outperforms the other two methods. I am not sure which other, fairer, evaluation criterion could be introduced, but I think that this issue should be addressed or at least thoroughly discussed.

> We have added the following section to clarify that, indeed, all three methods aim at minimising the difference between simulated and real climate, but make different assumptions as to how this aim can best be achieved:

>> All three bias correction methods considered here aim at minimising biases in past simulated data, but they make different assumptions as to how this aim can best be achieved. The Delta Method assumes that the (known) present-day model bias is also a suitable estimate for past model bias. GAM methods and Quantile Mapping operate on the premise that this assumption of that Delta Method - local biases remaining constant over time - is too strong. Instead, GAM methods assume that a better estimate of past model biases can be obtained by deriving a statistical relationship between present-day bias and present-day simulations, and then applying this relationship to past simulations in order to estimate past bias. By the nature of regression models, GAM methods do not perfectly explain present-day model biases across grid cells in terms of its predictor variables. As a result, and unlike in the case of the Delta Method, GAM-corrected present-day simulations are not identical to the present-day observed climate. This drawback is accepted under the assumption that the derived statistical model captures the *mechanisms* underlying local model biases better than the time-constant local correction term used in the Delta Method, and indeed to an extent that allows better estimates of past model biases. Similarly, Quantile Mapping assumes that the distributional correction of climate quantiles - whilst, again, not perfectly eliminating biases in present-day simulations - ultimately represents a better strategy for minimising past bias than the rigid local correction of the Delta Method.

> Although the Delta Method fully eliminates present-day bias, as pointed out by the Reviewer, a priori, it is not clear whether it would also reduce *past* biases most effectively. Indeed, our analysis demonstrates that this is not the case in several scenarios, which supports the rationale underlying both the GAM Method and Quantile Mapping, i.e. present-day bias is *not* as good an estimate for past bias as the one obtained by using these other two methods.

In our revised version, we begin our results section by providing plots showing, for each climate variable, point in time and bias correction method, the complete, unprocessed set of local biases, thus illustrating the performance of each method across the full spectrum of values of the relevant climate variable. Only after that do we present the statistical summary of these plots, in terms of the median absolute biases.

We now also motivate our evaluation approach in greater detail by means of the following new paragraph

> In ecological applications, the objective of applying a bias-correction method to past simulated climate data is generally to reduce the difference between the simulated and the (generally unknown) true past climate. Empirical palaeoclimatic reconstructions allow us to assess the differences at specific locations and points in time. Here, we determine these local differences between empirical reconstructions and bias-corrected simulations for each climate variable and bias-correction method, and define a spatially aggregated measure to assess the overall global performance of each method. [...] We provide complete plots of the distribution of the biases corresponding to each specific climate variable, point in time, and bias correction method. As a summary statistic of these distributions, and an aggregated measure for evaluating and comparing the performance of the three bias correction methods, we use the [MAB].

We would argue that the MAB is the most natural and intuitive way to statistically summarise the set of local biases, providing a simple measure to assess, as we state later on in the text, whether a bias-correction correction method overall improves the raw simulation outputs (namely if the associated MAB is smaller than that of the non-bias-corrected simulations).

In addition, following Reviewer 2's suggestion, we now additionally evaluate the performance of each method in terms of improving the simulated climate change signal, and have summarised these results in a newly added figure.

2) The difference between the corrected simulated climatologies and the reconstructed climatologies does not take into account the presumably large uncertainty in the reconstructions and in the corrected simulated climatologies (the former being presumably much larger?) . This needs to be incorporated in the evaluation of the three methods. If the inter-methodological differences are much smaller than the uncertainties in the estimated paleo-bias , it would be difficult to claim that one particular method is superior to other two. I think that the manuscript should include also these uncertainty estimations, or at least place the inter-methodological differences in the frame of the reconstruction uncertainties.

> We have added the following paragraph to the Methods:

> We tested whether the median absolute biases associated with two bias-correction methods, a specific climate variable and point in time, were significantly different, under the given uncertainty in the empirical reconstructions, using the following approach. For each climate variable and point in time, we generated $10^4$ Monte Carlo realisations of empirical past climatic values in the locations where reconstructions are available by

applying a normally-distributed noise term, with mean zero and standard deviation equal to the error of the local empirical reconstruction, to the value provided by the empirical reconstruction. Next, we calculated the local absolute biases between these empirical past climatic values, and the appropriate simulated values obtained after applying the different bias-correction methods. For each of these $10^4$ sets of absolute biases between empirical and simulated data, we used a one-sided Wilcoxon rank sum test to assess whether the median of the absolute biases associated with one bias-correction method was significantly smaller than that associated with a different bias-correction method (at a 5% significance level). We then determined the number of iterations, out of the total $10^4$ Monte Carlo realisations, in which this was the case. If, for a given climate variable and point in time, a bias-correction method was found to perform significantly better than another one in more than half of the realisations, we report this result in section 3.

and have added the following paragraph to the results:

These trends in the performances of the different bias-correction methods in terms of the median absolute bias are not always statistically significant (Fig. 2). The median absolute bias associated with the Delta Method was significantly smaller ($p < 0.05$) than that associated with Quantile Mapping and the GAM method for Mid-Holocene terrestrial mean annual temperature (in 96% and 83% of Monte Carlo realisations (see section 2.3) when compared against Quantile Mapping and the GAM method, respectively), marine mean annual temperature (in 93% and 89% of realisations, respectively), terrestrial mean temperature of the warmest month (in 92% and 100% of realisations, respectively), and precipitation (in 100% and 100% of realisations, respectively). The Delta Method also performed significantly better than the GAM method for Mid-Holocene terrestrial mean temperature of the coldest month (86% of realisations), and significantly better than Quantile mapping for LGM marine mean annual temperature (65% of realisations). The GAM method performed significantly better than Quantile mapping for LGM precipitation (100% of realisations). By construction, the Delta Method has a significantly lower median absolute bias (namely zero) than the other methods for all variables at present day.

We have also highlighted these results in Fig. 2, and have added caveats in the Discussion stating that the slightly better overall performance of the Delta Method in several cases is not always statistically significant.

3) The readability of the illustrations is poor. it is, for instance, very difficult to discern anything in Figure 2 and Figure 3. The lettering, axis labels, etc, in most figures is too small (e.g Figure 4)

We have increased the resolution and size of the maps. We have increased the font sizes in all figures.

4) what is the original spatial resolution of the climate reconstructions ? were they regridded, and how?

We have specified section 2.1.2 as follows:

Terrestrial temperature and precipitation reconstructions for the Mid-Holocene and the LGM are provided on a 2° resolution grid, and LGM marine temperature reconstructions are provided on a 5° grid. We assigned each sample of these datasets to the 1.25°x0.8° grid cell of our palaeoclimate simulations (see section 2.1.1) that contains the centre of the relevant 2° or 5° cell. Reconstructions for the Last Interglacial Period are not gridded, and were compared to the simulated climate in the 1.25°x0.8° grid cell containing the sample location. Fig. 3 and Fig. 4 visualise the locations of all reconstructions of terrestrial and marine mean annual temperature, and of annual precipitation.

5) The text refers sometimes to bias , other times to 'error', whereas in my understanding very often both terms carry the same meaning. This can be confusing for some readers. I would recommend to stick to one of those terms when possible.

We have removed the term 'error' as suggested, and now use 'bias' throughout the text.

6) The text also refers to the climate reconstructions as 'the observations', e-g. in equation 5. This can also be confusing. It would be clearer to use 'climate reconstructions' when referring to the proxy-reconstructed climatologies and 'observations' when referring to present-day climatologie.

We now use the terminology suggested by the Reviewer throughout the text.

7) The main conclusion is derived from the analysis of only one model. Perhaps I missed it but I think this a caveat that should be mentioned.

We have added this caveat to section 2.1.1, as suggested.

---

## Author Comment (AC2) · 17 Oct 2019

**Reviewer 2**

1) Page 1, lines 11-13, The DC method has its name because it is based on adding the simulated difference between two periods to observations. Although this is mathematically equivalent to subtracting the fitting period bias from the simulations (as shown in eqn.1) I think introducing the BC methods using the second definition rather than the one that is directly linked to the name is potentially confusing.

> We now introduce the Delta Method as follows:

>> The delta method is based on adding the difference between past and present-day simulated climate (the 'delta') to present-day observed climate.

> We have also changed the order of equations in Eqs. (1) and (2), as suggested by the Reviewer.

2) Page 1, lines 13 -15, Please use clean terminology. GAM is a statistical representation of links between 'variables' not between 'proxies for processes' and 'biases'. Define clearly what predictors and predictands are. From the current statement it is impossible to find out which variables are actually linked through GAM.

> We have rephrased the statement as follows:

>> GAMs attempt to represent statistical relationships between simulated climatic variables (as well as other known physical variables, such as elevation and the distance from the coast) and bias-corrected climatic variables (Vrac et al., 2007; Maraun and Widmann, 2018).

> We have also rewritten section 2.2.2, in which GAM methods are explained in detail (see our responses to comments further below).

3) Page 1, lines 15-16, QM does not assume the shape of the distribution to be constant in time. If there is climate change the distribution obviously changes. Standard implementations assume that the bias for a given value is constant in time (but there are implementations without this assumption). Please remove wrong statement and include a correct explanation.

> We have corrected the statement as follows:

>> Quantile mapping assumes that biases are specific to their respective quantiles in the distribution of the relevant climatic variable.

4) Page 1, lines 18-20. Please be more specific about the potential setups in the palaeoclimate context. Some empirical palaeoclimate reconstructions are local or have a high spatial resolution, which means they are smaller-scale than the climate model output (downscaling), whereas continental-scale empirical reconstructions have a lower resolution than the models (upscaling).

> We rewrote section 2.1.2 as as follows, to accommodate the Reviewer's comment:

>> We used global datasets of local palaeoclimate reconstructions of

terrestrial mean annual temperature, temperature of the coldest and warmest month, and annual precipitation for the mid-Holocene and the LGM from Bartlein et al. (2011), reconstructions of mean annual sea surface temperature for the mid-Holocene and the LGM from Hessler et al. (2014) and Waelbroeck et al. (2009), respectively, and reconstructions of mean annual continental and sea surface temperature for the last interglacial period from Turney and Jones (2010). Standard errors of reconstructed values are available for all variables with the exception of Last Interglacial terrestrial and marine temperature.

Terrestrial temperature and precipitation reconstructions for the Mid-Holocene and the LGM are provided on a 2° resolution grid, and LGM marine temperature reconstructions are provided on a 5° grid. We assigned each sample of these datasets to the 1.25°x0.8° grid cell of our palaeoclimate simulations (see section 2.1.1) that contains the centre of the relevant 2° or 5° cell. Reconstructions for the Last Interglacial Period are not gridded, and were compared to the simulated climate in the 1.25°x0.8° grid cell containing the sample location. Fig. 3 and Fig. 4 visualise the locations of all reconstructions of terrestrial and marine mean annual temperature, and of annual precipitation.

5) Page 1, There is a complete lack of critical discussion about the limitations of BC. It is obvious that a fundamentally poor model cannot be improved in a meaningful way by BC (see for instance Maraun and Widmam (2018), Maraun et al., 2017: Towards process informed bias correction of climate change simulations. Nature Climate Change, 7(11), 764-773). These limitations should be discussed in the introduction, in particular in the context of palaeoclimate simulations. Moreover, the validation approach needs to be justified taking into account the potential problems with BC, and clear comments need to be made on whether the validation would identify such problems. It will turn out that it would not (see comments below), which should at least be stated as a limitation of the study.

We have added the following paragraph to the Introduction:

Several challenges of methods used for bias-correcting future climate simulation data, including the correct representation of distributions of extreme weather events (e.g. precipitation during El Niño events, or dry spell lengths), of very small-scale patterns, or of the variability of climatic variables across time scales of a few years or decades (Maraun et al., 2017), are oftentimes not present in the paleoclimatological context. This is because palaeoclimate data is most often provided at a medium-scale spatial resolution, and represents millennial-scale averages. However, in both scenarios it is important to acknowledge that bias-correction methods are unable to substantially improve a fundamentally poor climate model, e.g. with strong circulation biases that such methods are not capable of removing (Maraun et al., 2017). Seeking to improve the representation of climate dynamics in simulation models therefore remains a priority alongside the development of bias correction methods.

In addition, we have added the following paragraph to the Introduction:

[H]ere, we focus on the global performance of the different methods; however, we note that bias-correction is not a one-size-fits-all approach

(Maraun et al., 2017), and that our results do not remove the need for local re-evaluations of methods in specific continental and subcontinental regions of interest.

and the following sentence to the Conclusion:

Given the substantial variability of the effectiveness of the different methods in different locations, we echo earlier propositions that studies focussing on specific regions require case-by-case assessments of which bias-correction method is most suitable for improving palaeoclimate model outputs (Maraun et al., 2017).

We have also added the following caveat to the definition of the MAB (previously MAE):

We emphasise that the MAB is a summary statistic of the degree to which a given bias-correction method reduces the difference between simulated and empirical climatic data of a specific type, i.e. it does not allow inference of the goodness of the climate model, or of the performance of each method in improving the representation of climatic signals that are not captured by the empirical data used here.

6) Page 4, lines 18-19, The statement about the log-transform is correct, but overcomplicated. It is more helpful to just say that this is a multiplicative delta method, i.e. the simulated relative change is applied to the observations.

We have removed the sentence referring to the log-transformation, as suggested, and have added the following statement:

This corresponds to applying the simulated relative change to the observations.

7) Page 4, lines 21-22, The sentence doesn't work out. The relationship is between climate model output and real-world climate variables, with additional time-invariant predictors such as topography or distance from the coast.

We have rewritten the sentence to clarify dependent and independent variables:

Statistical bias correction methods assume the existence of a functional relationship between (i) true climatic conditions (dependent variables), and (ii) climate model outputs as well as additional known forcings such as topography (independent variables) (Vrac et al., 2007; Maraun and Widmann, 2018). "

8) Page 5, eqn. 3, Clarify that some $x_i$ are time-dependent (i.e. those that represent climate model output), while others (topography, distance from coast) are not.

We now explicitly state the temporal dependency of the predictor variables in the equations, and have specified in the text that these are

time-dependent; not only when they represent climate model outputs, but also when they represent elevation or the distance to the ocean, which

vary over time as the result of sea level changes.

9) Page 5, line 13, does 'wind speed' include the direction?

We have added "(absolute)" to clarify that we mean speed, not velocity.

10) Page 5, line 14-15, It is not clear what the predictor and predictand data are and how the fitting for the $f_i$ works. What are the individual realisations of $T_{sim}$ and $x_i$ for which the polynomials are fitted? Are these timesteps? But if so, if I understand correctly, there are only three, namely the mean temperatures for the present, Mid-Holocene, and Last Glacial Maximum. What is the spatial resolution? Are the simulated temperatures averaged over the continental areas represented in the proxy-based reconstructions? Or are the realisations in space (if so, is this one value for each continent?), or space and time?

In our initial submission, we had abused mathematical notation in some instances (e.g. by dropping the dependence of certain variables on time, location, or the climate variable or bias correction method in question) with the aim of facilitating an intuitive understanding of the key concepts. We understand that this may have caused misunderstandings and loss of clarity of our methods. We have therefore completely rewritten the mathematical parts of section 2.2 (bias correction methods) and section 2.3 (method evaluation). We have explicitly added the dependence of variables on time, location, climate variable, and bias-correction method throughout these sections, thus clarifying the details that the Reviewer enquired about.

11) Page 5, line 17-22, It is not clear what the distributions are. Are they annual values of continental means?

In the course of rewriting the technical details of the methods (see response to previous comment), we have clarified the data that the relevant cumulative distribution are based on.

12) Page 6, The evaluation method needs more justification. For instance it would be a logical first step to validate the three BC methods on instrumental data, using crossvalidation, and focusing on aspect that are important in the palaeoclimatic context, i.e. long-term variability. The argument is probably that the key aspect is the representation of changes on multi-millennial timescales. The evaluation section should start with stating the objectives of the evaluations, followed by a justification of why the chosen evaluation method addresses these objectives. Please keep in mind that BC methods reduce bias by construction, even for completely wrong models (see e.g. Maraun et al, 2017). A reduction in the bias of the mean (DC, GAM), will reduce also the biases for the distribution quantiles, while BC corrects these directly. For strongly biased climate models the reduction of the biases in the distributions will necessarily lead to a reduction in MAE. Why is the MAE chosen as the evaluation measure? In the paleoclimate context it is also very relevant to compare the climate change signals in the raw and the BC-corrected simulations, and in the proxybased reconstructions. Please add statements and if suitable figures on this.

We have added the following paragraph to the beginning of the section 2.3 (Model evaluation) to clarify the objective of our evaluation:

In ecological applications, the objective of applying a bias-correction method to past simulated climate data is generally to reduce the difference between the simulated and the (generally unknown) true past climate. Empirical palaeoclimatic reconstructions allow us to assess the differences at specific locations and points in time. Here, we determine these local differences between empirical reconstructions and bias-corrected simulations for each climate variable and bias-correction method, and define a spatially aggregated measure to assess the overall global performance of each method.

After formally defining the local differences between empirically reconstructed and bias-corrected simulated data, we motivate the use of the MAB as follows:

We provide complete plots of the distribution of the biases corresponding to each specific climate variable, point in time, and bias correction method. As a summary statistic of these distributions, and an aggregated measure for evaluating and comparing the performance of the three bias correction methods, we use the [MAB].

We would argue that the MAB is the most natural and intuitive way to statistically summarise the set of local biases, providing a simple measure to assess, as we state later on in the text, whether a bias-correction correction method overall improves the raw simulation outputs (namely if the associated MAB is smaller than that of the non-bias-corrected simulations).

However, we have added Figs.1a-e, showing for each climate variable, point in time and bias correction method, the unprocessed complete set of local biases, thus illustrating the performance of each method across the full spectrum of values of the relevant climate variable. We only show the statistical summary of these plots, in terms of the MAB, in Fig. 2.

We agree with the Reviewer that bias-correction methods reduce the overall bias in present-day simulations, and we now explicitly state this in the text. However, we would argue that it is not clear, a priori, whether any of the three bias-correction methods considered also reduces biases in past simulations. Indeed, our analysis shows that this is not always the case: Some bias-corrected simulations have a higher MAB than the raw simulation data.

As suggested by the Reviewer, we have included an evaluation of the performance of each bias-correction method in terms of reducing the average bias between the empirically reconstructed and the simulated climate change signal, which may be relevant in certain applications. We have added the formal details of this evaluation to section 2.3 (Model evaluation). A newly added figure shows that the differences between the methods in terms of bias-correcting the climate change signal are extremely small.

13) Page 5, line 7, 'standard errors' of what? It is said later that it is the error of the reconstructions, but it needs to be said the first time this is mentioned.

We have added information on the standard errors of the empirical data to the description of the empirical reconstructions in section 2.1.2.

14) Page 6, eqn. 6, The notation is very unclear. It is also not clear what 'grid cell' refers to. Earlier it was mentioned that continental means are used. This problem is related to the lack of clarity about predictand and predictor data mentioned in previous comments.

> As mentioned in our response to a previous comment by the Reviewer, we have completely rewritten and clarified the mathematical parts of our methods. This includes the section referred to by the Reviewer.
>
> We feel that the term "continental", which we have used in the sense of "terrestrial" (e.g. like Bartlein et al. (2011), our source of Mid-Holocene and LGM empirical reconstructions), may have led to confusion about the spatial scale of the empirical reconstructions used in our analysis. These are always local/gridded, never spatially aggregated across continents. (Thus, "Continental mean annual temperature" referred to the (locally specific) mean annual temperature of terrestrial data points.) We have clarified this in our methods by emphasising the locality and spatial dependence of variables. In addition, we now use the term "terrestrial" instead of "continental" throughout the text.

15) Page 7, figure 1. It seems not plausible that QM leads to substantially larger MAEs than for the raw simulations, with values up to 10 K. Surprisingly this is not even discussed. There might be a problem with the implementation. If the implementation is correct, please give a detailed explanation how this is possible. If I understand correctly the BCcorrected distribution of the present simulation is identical to the distribution of the instrumental observations. This means that the instrumental observations have also a very high MAE for the present. How can this be the case? If suitable, please add information about how the instrumental data, which are the training data for all BC methods, perform in this evaluation framework. When addressing this please state explicitly what is compared with what for calculating the MAE; the information that is currently given is incomplete.

> There was indeed an error in the implementation of Quantile Mapping. We have corrected this error, and find that Quantile Mapping also slightly reduces model biases, as expected by the Reviewer. We have updated the figures and text accordingly.
>
> The Reviewer is correct in that the cumulative distribution function of present-day simulated climatic values obtained after applying Quantile Mapping is identical to the cumulative distribution function of present-day observed values. However, this does not imply that the underlying climate maps must be identical (in which case the MAB would be 0). Indeed, any spatial permutation of present-day observed climate values would have the same cumulative distribution function as present-day observed climate, but the MAB would not necessarily be 0. Only in the case of the Delta Method are present-day observed climate and bias-correct simulated data identical (as are, by extension, their cumulative distribution functions).

16) Page 11, figure 4. If I understand correctly, this figure shows that the simulated climate change signal is different from the reconstructed climate change signal. If this is correct, please include this straightforward interpretation.

> This is not the case. Letting $V_{sim}(x,t)$ and $V_{emp}(x,t)$ denote the simulated and empirical values of a climate variable V at time t and location x. The figures suggest a relationship between "Past minus present model bias" - i.e. $(V_{emp}(x,t)-V_{sim}(x,t)) - (V_{emp}(x,0)-V_{sim}(x,0))$ - on the one hand, and the "Simulated climate

change" - i.e. $V_{sim}(x,t) - V_{sim}(x,0)$ - on the other hand. This is different from the Reviewer's suggestion that "the simulated climate change signal ", $V_{sim}(x,t) - V_{sim}(x,0)$, "is different from the reconstructed climate change signal", $V_{emp}(x,t) - V_{emp}(x,0)$.

---

## Author Response (AR2)

We are grateful to the Reviewers for their helpful comments, which we have accommodated as detailed in the point-by-point response below. We have also clarified the wording throughout the text, and reordered a few sentences, in order to improve the overall readability.

**Reviewer 1**

I thank the authors for having taken into consideration my concerns on the previous version. To my mind this version is very close to a publishable form have just three minor suggestions: in the introduction, the authors mention the order of magnitude of typical model bias: degrees for temperature, cm for annual precipitation. Since the delta method uses a multiplicative correction for precipitation, I would add also a typical percentage ( or range of it) of precipitation bias.

As can be seen from the below Figure 9.4d of the latest IPCC report, present-day relative precipitation biases can range between -100% to over +100%, but vary considerably across space, making it difficult to define a 'typical percentage'.

[Figure]

To accommodate the Reviewer's suggestion, we have changed the sentence to

"these biases can oftentimes be of the order of several degrees of temperature, or tens of percent of precipitation.

In the conclusions, the authors conclude that the Delta method tends to perform better than the other wo. Again, I would mention here again that the results (may) depend on the climate model used. Most readers will just read the conclusions and infer that this conclusion has overall validity, which I think it has not been shown in this study.

We have added the following statement to the Conclusions:

We also reiterate that our results may be different for palaeoclimate simulations other than the ones used here.

Figures 3 and 4 are still difficult to read. I am aware that it is hard to convey the information the authors wish to. I would urge the authors to spend some time thinking about alternatives. One that they could try is to use discreet, instead of continuous, colour coding, but I am not convinced that this will improve the figures

We have changed the colouring to a discrete scale, as suggested, and have increased the resolution and overall quality of Figs. 3 and 4. We feel that the key aim of the figures,

to visualise major spatial clusters where the bias correction performance is particularly good or poor, as well as to highlight the overall spatial heterogeneity in the performance, is achieved well with the current format without requiring excessive manuscript space.

**Reviewer 2**

First, a clear discussion of the kinds of biases the authors are targeting should appear at the beginning of the paper. The examples of distributions of extreme weather events or climate variability in the introduction seem to be a bit of a red herring, as those "biases" have more to do with higher moments of probability distribution than the the time-mean biases that are ultimately the focus of the paper.

> We have replaced the passage pointed out by the Reviewer by the following paragraph:

>> In many of these applications, climatological normals at quasi-equilibrium of variables such as temperature and precipitation at different points in time represent the most relevant climatic inputs. [...] Three main methods have been used to bias-correct climatological normals in the palaeocontext: [...] Here we combine a set of high-resolution simulations of the climatological means [...]

> We moved part of the replaced passage to the Conclusion.

Second, the absence of a discussion of paleoclimate state estimation and data assimilation is conspicuous given that the goals of those procedures are also to reduce misfits between models and paleo data. A starting point is the literature on offline Kalman filtering (e.g., Tardif et al. 2019, Clim Past, Last Millennium Reanalysis with an expanded proxy database and seasonal proxy modeling), particle filtering (e.g., Goosse 2016, Clim. Dyn., Reconstructed and simulated temperature asymmetry between continents in both hemispheres over the last centuries) and "online" state estimation that changes model forcing to generate new runs (e.g. Kurahashi-Nakamura et al. 2017, Paleoceanography, Dynamical reconstruction of the global ocean state during the Last Glacial Maximum and Amrhein et al. 2018, J. Clim, A Global Glacial Ocean State Estimate Constrained by Upper-Ocean Temperature Proxies). A comparison and discussion of complementarity would strengthen the paper and make it more relevant to CoP readers.

> We have added the following paragraph to our discussion of Fig. 5.

>> Such an approach would tie in with data assimilation methods, which also use empirical climate proxy records to improve climate simulations. These methods have been used to estimate global climate variables at times at which the quantity and spatial coverage of available empirical records is high enough to allow a robust calibration of the relevant computational methods. As a result, they have focussed either on single points in the past, such as the Mid-Holocene (Mairesse et al. 2013) or Last Glacial Maximum (Kurahashi-Nakamura et al. 2017), or on time intervals across which suitable empirical data are available, namely the last millennium period (Tardif et al. 2019, Gosset 2017). In contrast to the aforementioned approaches, based on Fig. 5 we suggest that it may be possible to use empirical reconstructions even from only a small set of points in time (e.g. the present, Mid-Holocene, LGM and Last Interglacial Period) to parameterise a statistical model of the temporal variation of local biases that could be used to improve simulated data at any time point in the Late Pleistocene-Holocene period.

p1l8-11 "slightly better…methods" It sounds like a more apt description is that the methods are indistinguishable. I would clarify what is meant by "slightly better"

> We have added the following statement to the Abstract:
>
>> In most cases, the differences between the bias reductions achieved by the three methods are small.
>
> We would not agree with the conclusion that our findings show that the methods are indistinguishable. As we report in section 3, the Delta Method leads to significantly smaller median absolute biases for a number of variables and points in time than the other two methods, whereas only in 2 out of the 17 total scenarios displayed in Fig. 2 is the median absolute bias not smallest for the Delta Method.
> We agree that the relative differences between the bias reductions achieved by the different methods are not always large, and therefore chose the phrasing "overall [...] performs slightly better" in the abstract. We feel that the following caveats "albeit not always to a statistically significant degree" and "however, there is considerable spatial and temporal variation in the performance of each of the three methods" add nuance to the statement with the brevity appropriate for the abstract.

p1l11 Should be a semicolon before however

> We have added a semicolon.

p1l11 Please clarify what is meant by reconstructions — data? Potentially confusing because reconstructions often use model output. Please also comment on the utility of using interpolated products (e.g. the MARGO gridded product) to evaluate bias reduction, as those products have their own (likely biased) assumptions of spatiotemporal covariance built in.

> We have revised the text and now use the term "reconstruction" exclusively in the context of data derived from empirical records. We agree that empirical climate reconstructions can themselves be subject to biases. This applies to the MARGO dataset, mentioned by the reviewer, as it does to pollen-derived reconstructions, where the transfer functions employed can be biased and subject to uncertainty. We have added the following statement to section 2.1.2 of the Methods accommodate the Reviewer's comment:
>
>> Empirically derived climate reconstructions can themselves be subject to biases and uncertainties, which arise at the different stages of the reconstruction process, from collecting the data to computationally converting empirical records to climatic variables. Nonetheless, these data represent the best empirically-based estimates of past climatic conditions available, and the most suitable data for our analysis.

p1l12 Please define what is meant by "active calibration" and "bias correction functions"

> We have rephrased the sentence as follows:
>
>> Our data also indicate that it could soon be possible to use empirical reconstructions of past climatic conditions not only for the evaluation of bias correction methods, but for fitting statistical relationships between empirical and simulated data through time that can inform more effective bias correction methods.

We have also added a model example to our description of Fig. 5 to illustrate a possible way to use past empirical reconstructions to improve the Delta Method.

p2l12 Please clarify what is meant by medium-scale. More accurate than "millennial-scale averages" might be "quasi-equilibrated climate states" when models are run for millennia. But I would dispute that these issues are not present in paleoclimate studies (e.g., the paleo drought literature).

We removed the sentence containing the phrase "medium-scale" in the course of addressing the Reviewer's second comment (see above). We now clarify in section 2.1.1. of the Methods that the climate simulations used in our analysis represent climatological normals at quasi-equilibrium, i.e. following a 500-year spin-up period.

p2l27 "Finally…" This sentence needs clarification.

We have rephrased the sentence as follows:

Quantile Mapping adjusts the cumulative distribution of the simulated data by applying a transformation between the quantiles of present-day simulated and observed climate to the quantiles of past simulated climate.

p2l28 "However…" What about the common practice of comparing paleoclimate anomalies in models and data? Isn't that a validation of the "Delta method"? e.g., Brady et al. 2013, J. Cli., Sensitivity to Glacial Forcing in the CCSM4.

We agree with the Reviewer that it is not uncommon to compare paleoclimate simulation output to empirically derived reconstructions (cf. Fig. 9.11 from IPCC AR5 WG1); however, this is different from comparing the performance of alternative bias correction methods, which is the aim of our work. To avoid confusion, and because the sentence pointed out by the Reviewer is not essential to the argument, we have removed it.

p3l16 Please provide more detail on the model simulations. Were they run to equilibrium? Biases can emerge when models are run for long periods of time (Amrhein et al. 2018, cited above), but long runs are also necessary to equilibrate climate states to forcings (particularly in the deep ocean, e.g. Jansen et al. 2018 J. Cli., Transient versus Equilibrium Response of the Ocean's Overturning Circulation to Warming).

We have clarified that the climate simulations used in our analysis represent climatological normals at quasi-equilibrium, i.e. after a 500-year spin-up period.

p3l19 It appears that the reference for the Last Interglacial has not been published. I'm not sure what Clim. Past's policy is here, but it's difficult to evaluate that output.

We leave it to the discretion of the Editor to decide whether referencing data that is presented in a preprint is acceptable.

p6l23 Please define "distributional bias" and "quantile." This introductory paragraph (and the rest of the section) are difficult to to understand. What CDFs are being discussed? Perhaps the following paragraph (p7l3) should come first.

We have rewritten the paragraph as follows, and added an example to clarify the approach:

Quantile Mapping aims to correct distributional biases in the simulated climate data. The method consists of first computing a transformation that maps the quantiles of the cumulative distribution function of all present-day observed values (i.e. from all land or ocean grid cells) of a climate variable onto the quantiles of the cumulative distribution function of all present-day simulated values. The derived mapping is then applied to the cumulative distribution function of all simulated values at a given point in the past. For example, let the cumulative distribution function of the values of present-day observed terrestrial mean annual temperature (i.e. from all land grid cells) map the value $T_1$°C onto the value $q \in [0,1]$, and let the analogous cumulative distribution function of present-day simulated terrestrial mean annual temperature map $T_2$°C onto q. If the value that is mapped onto q by the cumulative distribution function of simulated terrestrial mean annual temperature at a given point in the past is $T_3$°C, then the bias-corrected mean annual temperature in all grid cells with simulated mean annual temperature $T_3$°C at that point in time is estimated as $T_3$ + ($T_1$ - $T_2$) °C. Notably, by design of the method, after applying Quantile Mapping, the cumulative distribution function of present-day bias-corrected simulated data is identical to the cumulative distribution functions of present-day observed values.

In the following paragraph, we have clarified that V represents one of the five climatic variables considered in our analysis. We have also clarified the domain of the cumulative distribution functions.

We feel that knowledge of the analytical details of each of the different bias correction methods developed in sections 2.2.1 - 2.2.3 is required before the methods can be placed into context and discussed in section 2.2.4. We have therefore not changed the order of sections.

p7l21 "By the nature of regression models" -- unclear what is meant here, please clarify

We have rephrase the relevant phase as follows:

Because regressions generally do not fit the data perfectly, present-day biases modelled by the GAM will not exactly match the observed biases across all grid cells.

p8l9 measures -> measure

We have corrected the typo.

p8l12 I think that writing this out rather than using set notation would be more accessible to the readership of this journal.

We have removed the inaccessible notation, and rephrased the sentence as follows:

For a climate variable V (representing the relevant temperature and precipitation variables considered here) ...

p10l4 Please define the difference between median bias and median absolute bias.

We have added "(Eq. 7)" and "(Eq. 8)" to refer to the definitions of the two statistics.

Figure 1 Why are error bars only on a subset of the data?

Error bars for temperature data from the Last Interglacial Period are not available to us. The supplementary material of the relevant paper (from 2010) only contains the estimated values, but not uncertainties. We had contacted the authors of the paper at an earlier point in time, but did not receive a response. We have stated in the Methods: "
[revised manuscript text omitted]
_{\mathrm{sim}}^{raw}(x_1, t), V_{\mathrm{emp}}^{raw}(x_2, t), \dots$ at time $t$. We denote by

195  $F_{\mathrm{emp}}^V[0]^{-1}$ and $F_{\mathrm{sim}}^{V,raw}[t]^{-1}$ (both mapping $[0,1]$ to $\mathbb{R}$) the inverse functions of $F_{\mathrm{emp}}^V[0]$ and $F_{\mathrm{sim}}^{V,raw}[t]$, respectively.

With this notation, $F_{\mathrm{sim}}^{V,raw}[t](V_{\mathrm{sim}}^{raw}(x_i, t))$ is the quantile corresponding to the value $V_{\mathrm{sim}}^{raw}(x_i, t)$ in the set of  all simulated values of the climate variable $V$ at time $t$. Under Quantile Mapping, the function $\left[ \overline{F_{\mathrm{emp}}^V[0]^{-1} - F_{\mathrm{sim}}^{V,raw}[0]^{-1}} \right] F_{\mathrm{emp}}^V[0]^{-1} - F_{\mathrm{sim}}^{V,raw}[0]^{-1}$ maps each such quantile to a quantile-specific correction term, which is then applied to the raw simulation data. Thus,

200  $$V_{\mathrm{sim}}^{QM}(x_i, t) := V_{\mathrm{sim}}^{raw}(x_i, t) + \underbrace{\left[ F_{\mathrm{emp}}^V[0]^{-1} - F_{\mathrm{sim}}^{V,raw}[0]^{-1} \right] \left( F_{\mathrm{sim}}^{V,raw}[t](V_{\mathrm{sim}}^{raw}(x_i, t)) \right)}_{\text{Correction term specific to the quantile of } V_{\mathrm{sim}}^{raw}(x_i, t)}.$$

 the bias-corrected value of $V$ in the location $x_i$ at time $t$  is estimated as

$$V_{\mathrm{sim}}^{QM}(x_i, t) := V_{\mathrm{sim}}^{raw}(x_i, t) + \underbrace{\left[ F_{\mathrm{emp}}^V[0]^{-1} - F_{\mathrm{sim}}^{V,raw}[0]^{-1} \right] \left( F_{\mathrm{sim}}^{V,raw}[t](V_{\mathrm{sim}}^{raw}(x_i, t)) \right)}_{\
[revised manuscript text omitted]

---

## Author Response (AR3)

Dr Robert Beyer
Department of Zoology
University of Cambridge,
Cambridge CB2 3EJ
United Kingdom

[Figure]

UNIVERSITY OF
CAMBRIDGE
Department of Zoology

08 July 2020

Dear Dr Phipps,

We were please to receive your decision on our manuscript, and have attached the revised version, in which we corrected the typo and completed the Data and Code availability statement, as requested.

Best regards,

Robert Beyer

Dr Robert Beyer
Department of Zoology
University of Cambridge,
Cambridge CB2 3EJ
United Kingdom